# Learning few-step posterior samplers by unfolding and distillation of diffusion models

**Charlesquin Kemajou Mbakam**                                    *cmk2000@hw.ac.uk*
*School of Mathematical and Computer Sciences &*
*Maxwell Institute for Mathematical Sciences*
*Heriot-Watt University*

**Jonathan Spence**                                               *JSpence5@ed.ac.uk*
*School of Mathematics &*
*Maxwell Institute for Mathematical Sciences*
*University of Edinburgh*

**Marcelo Pereyra**                                               *M.Pereyra@hw.ac.uk*
*School of Mathematical and Computer Sciences &*
*Maxwell Institute for Mathematical Sciences*
*Heriot-Watt University*

**Reviewed on OpenReview:** *https://openreview.net/forum?id=oGCfD8YKN2*

## Abstract

Diffusion models (DMs) have emerged as powerful image priors in Bayesian computational imaging. Two primary strategies have been proposed for leveraging DMs in this context: Plug-and-Play methods, which are zero-shot and highly flexible but rely on approximations; and specialized conditional DMs, which achieve higher accuracy and faster inference for specific tasks through supervised training. In this work, we introduce a novel framework that integrates deep unfolding and model distillation to transform a DM image prior into a few-step conditional model for posterior sampling. A central innovation of our approach is the unfolding of a Markov chain Monte Carlo (MCMC) algorithm—specifically, the recently proposed LATINO Langevin sampler (Spagnoletti et al., 2025)—representing the first known instance of deep unfolding applied to a Monte Carlo sampling scheme. We demonstrate our proposed unfolded and distilled samplers through extensive experiments and comparisons with the state of the art, where they achieve excellent accuracy and computational efficiency, while retaining the flexibility to adapt to variations in the forward model at inference time. The code is available at https://github.com/charles-kmc/UD2M.

## 1 Introduction

We seek to infer an unknown image of interest $x^\star \in \mathbb{R}^d$ from a noisy measurement $\mathbf{y} = y$, related to $x^\star$ by a statistical observation model of the form

$$\mathbf{y} \sim P(\boldsymbol{A}x^\star)\,, \tag{1}$$

where $\boldsymbol{A}$ is a linear measurement operator describing deterministic instrumental aspects of the data acquisition process and $P$ is a statistical model describing measurement noise. This includes, for instance, the canonical linear Gaussian observation model $\mathbf{y} = \boldsymbol{A}x^\star + \mathbf{n}$, where $\mathbf{n}$ is additive Gaussian noise with covariance $\sigma_n^2 \boldsymbol{I}$. Such models arise frequently in problems related to image deblurring, inpainting, super-resolution, compressive sensing, and tomographic reconstruction (see, e.g., (Kaipio & Somersalo, 2010; Ongie et al., 2020)). Within this context, we are particularly interested in imaging problems where recovering $x^\star$ from $y$ is severely ill-conditioned or ill-posed, leading to significant uncertainty about the solution (Kaipio & Somersalo, 2010).

Adopting a Bayesian approach, we leverage prior knowledge available in order to regularize the problem and deliver meaningful inferences that are well-posed. This is achieved by treating $x^\star$ as a realization of a random variable $\mathbf{x}_0$ and using Bayes' theorem to obtain the posterior distribution of $\mathbf{x}_0|\mathbf{y} = y$, with density given for all $x_0 \in \mathbb{R}^d$ by (Robert, 2007)

$$p(x_0|y) = \frac{p(y|x_0)p(x_0)}{\int_{\mathbb{R}^d} p(y|\tilde{x}_0)p(\tilde{x}_0)\mathrm{d}\tilde{x}_0} \,,$$

where $p(y|x_0)$ denotes the likelihood function associated to Equation 1 and $p(x_0)$ the marginal density of $\mathbf{x}_0$.

The choice of $p(x_0)$, so-called prior, is crucial and greatly impacts the obtained posterior $p(x_0|y)$, especially in problems that are ill-conditioned or ill-posed. Bayesian imaging methods traditionally used analytical priors designed to promote solutions with expected structural properties (e.g., smoothness, sparsity, piecewise regularity). However, modern Bayesian imaging methods rely predominantly on machine learning in order to harness vast amounts of prior knowledge available in the form of training data and deliver unprecedented accuracy (Mukherjee et al., 2023). In particular, state-of-the-art Bayesian imaging techniques often use deep generative models as image priors, notably denoising diffusion models (DMs). For an introduction to DMs, we refer the reader to the seminal DM papers Song & Ermon (2019; 2020); Song et al. and to the recent survey on DMs as image priors for solving inverse problems (Daras et al., 2024).

The literature on DM-based Bayesian imaging methodology has two main strands. On the one hand, Plug & Play (PnP) methods seek to use a pretrained DM in a zero-shot manner, in combination with a likelihood function $p(y|x_0)$ specified analytically during inference. PnP approaches are flexible by construction and generalize robustly to new measurement models by exploiting knowledge of the likelihood function $p(y|x)$ explicitly. Notable examples of PnP DM techniques include, e.g., DPS (Chung et al., 2022), DDRM (Kawar et al., 2022), DiffPIR (Zhu et al., 2023), ΠGDM (Song et al., 2023a), and mid-point (Moufad et al., 2025).

The other strand relies on fine-tuning an unconditional DM representing the prior, in order to construct a $y$-conditional DM for posterior sampling. This involves modifying a DM to take the measurement $y$ as input and, through supervised training, specializing it for posterior sampling of $\mathbf{x}_0|\mathbf{y} = y$ (instead of marginal sampling of $\mathbf{x}_0$). Specializing a DM for a specific imaging task enables the development of Bayesian imaging methods that surpass PnP approaches in both accuracy and computational efficiency, assuming the task is fully defined during training. However, in the absence of additional training, such task-specific DMs often exhibit limited generalization to even slightly altered measurement conditions, as they are not inherently designed to exploit the likelihood $p(y|x_0)$ during inference. For more details and comparisons with PnP strategies, see, e.g., I$^2$SB (Liu et al., 2023a), InDI (Delbracio & Milanfar, 2023), CDDB (Chung et al., 2023).

Moreover, the most advanced DM-based imaging methods currently available -whether zero-shot or trained for conditional sampling- leverage model distillation to enhance computational efficiency and improve performance. Notably, distillation techniques such as consistency models (Song et al., 2023c; Kim et al., 2024) and flow matching (Liu et al., 2023b; Lipman et al., 2023) have significantly reduced the number of required neural function evaluations (NFEs) per posterior sample from over $10^3$ to fewer than 10, while concurrently improving sampling quality and accuracy (see Spagnoletti et al. (2025); Garber & Tirer (2025); Zhao et al. (2025)).

To bridge the methodological gap between these two strands, this paper introduces a novel framework for constructing conditional DMs which offers the computational efficiency and accuracy of conditional DMs obtained via finetuning and distillation, with the flexibility afforded by explicit likelihood functions in inference, thus bringing together the advantages of finetuning and PnP strategies. This is achieved by generalizing deep unfolding (Monga et al., 2021), a paradigm wherein iterative algorithms with a fixed number of steps are mapped onto modular deep neural network architectures. These architectures explicitly incorporate the likelihood function $p(y|x_0)$, for example through its gradient or proximal operator, in conjunction with a denoiser that is subsequently fine-tuned during training. A main novelty of this work is to present the first application of deep unfolding of a Markov chain Monte Carlo (MCMC) sampling scheme, in contrast to prior work on deep unfolding that focused on optimization-based algorithms (which, unlike MCMC schemes, are not inherently suitable for posterior sampling). Namely, we propose to unfold the state-of-the-art LATINO zero-shot imaging Langevin method (Spagnoletti et al., 2025), and train the resulting unfolded LATINO networks by using a supervised consistency trajectory models objective combining distortion, perceptual and adversarial terms (Kim et al., 2024). This leads to distilled conditional DMs that are fast (low NFEs),

|  | Deblurring | Inpainting | SR (×4) | JPEG (QF=10) |

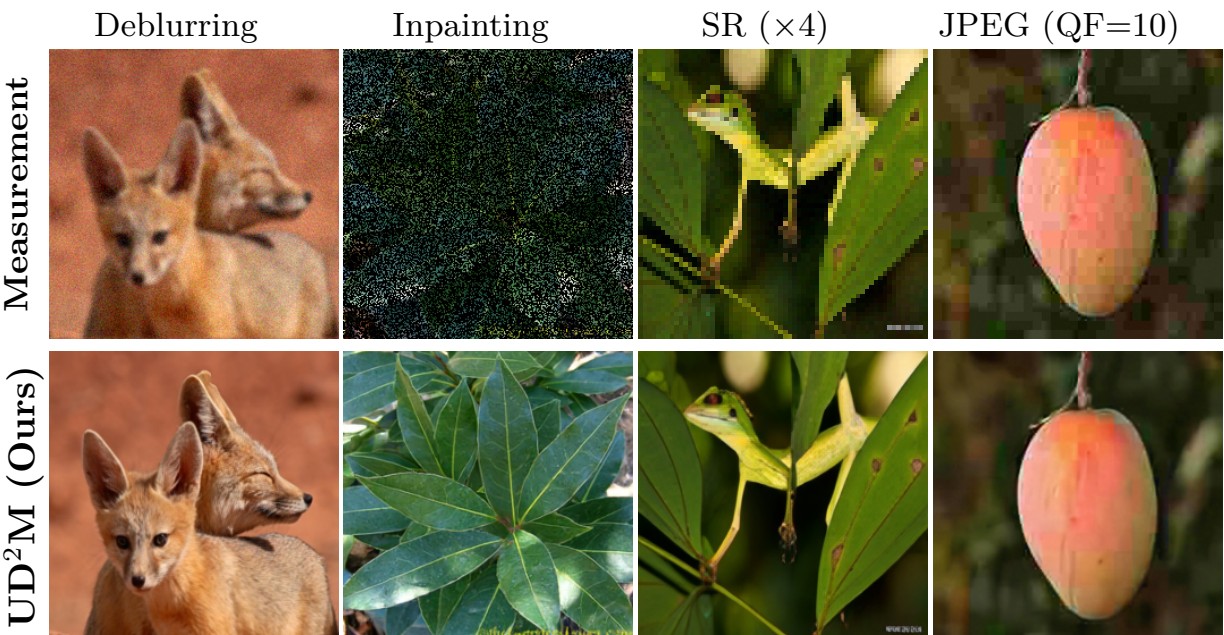

Figure 1: Qualitative comparison of the proposed Unfolded and Distilled Diffusion Model (UD²M) for posterior sampling on the ImageNet 256 dataset. Tasks: Gaussian Deblurring, random inpainting (70%), super-resolution (4×), and restoration of JPEG compression artifacts (QF=10).

generate high-quality samples, are GPU-memory efficient during inference (due to the absence of automatic differentiation). We refer to the proposed method as an Unfolded and Distilled Diffusion Model (UD²M). Crucially, UD²Ms support joint training over a family of likelihoods, followed by instantaneous specialization to specific measurement models at inference time (i.e., specific instances of $\boldsymbol{A}$ and $P$). For illustration, Figure 1 shows some examples of posterior samples obtained with the proposed unfolded and distilled diffusion models (UD²Ms) applied to Gaussian deblurring, random inpainting, super-resolution by a factor 4, and JPEG artifact restoration for images from ImageNet dataset.

In summary, we present the following two main methodological contributions that underpin the proposed UD²M method:

- Deep unfolding of Monte Carlo sampling: In Section 3, we introduce a novel framework that unfolds the LATINO Langevin Markov chain Monte Carlo sampler into a trainable neural architecture, representing the first application of deep unfolding to a Monte Carlo sampling scheme.

- Conditional model distillation for efficient posterior sampling: In Section 3.2, we employ modern generative model distillation techniques to fine-tune the unfolded LATINO network into a few-step conditional sampler that achieves high accuracy and flexibility while enabling fast inference.

A summary of background material can be found in Section 2 and numerical comparisons of our method to state-of-the-art models from the literature are presented in Section 4.

## 1.1 Notation

We denote by $p(x|y)$ the posterior distribution of the random variable $\mathbf{x}$ given $\mathbf{y} = y$. Similarly, $p(y|x)$ denotes the likelihood. We use the shortened notation $-\log p(y|x) = g_y(x)$. For a function $f : \mathbb{R}^d \to \mathbb{R}$, the operator $\text{prox}_{\delta f}(x) = \arg\min_{z \in \mathbb{R}^d} f(z) + \frac{1}{2\delta}\|x - z\|_2^2/2$ denotes the proximal operator of $f$. The expected value of a random variable $f(\mathbf{x})$ is denoted $\mathbb{E}_{\mathbf{x}}[f(\mathbf{x})]$; when the distribution to be integrated is clear from context, we occasionally drop the subscript and write $\mathbb{E}[f(\mathbf{x})]$.

## 2    Background & related works

**Diffusion models**    (DMs) are generative models that draw samples from a distribution of interest $p(\mathbf{x}_0)$ by iteratively reversing the following "noising" process

$$d\mathbf{x}_t = -\frac{\beta_t}{2}\mathbf{x}_t dt + \sqrt{\beta_t}d\mathbf{w}_t \,, \tag{2}$$

which is designed to transport $\mathbf{x}_0$ to a standard normal random variable $\mathbf{x}_T \sim \mathcal{N}(\mathbf{0}, \boldsymbol{I})$ as $T \to \infty$, and where $\beta_t$ represents a noise schedule and $\mathbf{w}_t$ is a Brownian motion (Ho et al., 2020). The reverse process, which allows generating samples of $\mathbf{x}_0$ by transporting $\mathbf{x}_T \sim \mathcal{N}(\mathbf{0}, \boldsymbol{I})$ back to $\mathbf{x}_0$, is given by (Song et al., 2021):

$$d\mathbf{x}_t = \left[-\frac{\beta_t}{2}\mathbf{x}_t - \beta_t \nabla_{x_t} \log p_t(\mathbf{x}_t)\right] dt + \sqrt{\beta_t}d\overline{\mathbf{w}_t}, \tag{3}$$

where $\overline{\mathbf{w}_t}$ is again Brownian motion, in the reverse direction. The target distribution $p(\mathbf{x}_0)$ is encoded in the so-called score function $x_t \mapsto \nabla_{x_t} \log p_t(x_t)$, which is approximated using Tweedie's identity (Efron, 2011) by a deep neural network denoiser $G_\theta(\cdot, t)$ that is trained by weighted denoising score matching (DSM), see Song et al. (2021).

Denoising diffusion probabilistic models (DDPM) (Ho et al., 2020) and denoising diffusion implicit models (DDIM) (Song et al., 2021) are the two main methodologies to implement Equation 3 in a time-discrete setting, where the process is initialized with $\mathbf{x}_T \sim \mathcal{N}(\mathbf{0}, \boldsymbol{I})$ for some large finite $T$ and solved iteratively with $t$ decreasing progressively to $t = 0$. Accurate estimation of the scores $\nabla_{x_t} \log p_t(x_t)$ underpinning Equation 3 has been crucial to the success of DDPM and DDIM. This has been achieved using large training datasets, specialized network architectures, and significant high-performance computing resources (Ho et al., 2020; Song et al., 2021). In addition to advancing generative modeling, DMs have become central to modern strategies for solving inverse problems in computational imaging (see, e.g., the recent survey paper (Daras et al., 2024)).

**Distillation of diffusion models**    Conventional DM approaches produce high quality samples by progressively transporting $\mathbf{x}_T$ to $\mathbf{x}_0$. However, they have a high computational cost, as generating each sample typically requires performing between 100 and 1000 evaluations of $G_\theta$. This drawback can be addressed by distilling $G_\theta$ into a new model $\tilde{G}_\vartheta$ requiring far fewer steps (e.g., 4-8 steps); see, e.g., Song et al. (2023c); Liu et al. (2023b). Remarkably, modern distillation techniques combining DSM with adversarial training are able to further improve the quality of the samples as they dramatically reduce the number of sampling steps, while retaining the same network architecture as the original DM. This is the case, for example, of the so-called consistency trajectory models (Kim et al., 2023), which we use in our proposed method in Section 4.

**Consistency models**    Consistency models (CMs) are derived from a probability flow formulation of Equation 3, given by (Song et al., 2023b)

$$dx_t = \left[-\frac{\beta_t}{2}x_t - \frac{\beta_t}{2}\nabla_{x_t} \log p_t(x_t)\right] dt. \tag{4}$$

This ODE is equivalent to Equation 3 in the sense that both have the same marginals $p_t(\cdot)$ of $\mathbf{x}_t$. CMs leverage this property by learning a so-called consistency function $\tilde{G}_\theta$ that maps any point $x_t$ on a trajectory $\{x_t\}_{t \in [\eta, T]}$ of Equation 4 backwards to $x_\eta$, for some small given $\eta > 0$. Given $\tilde{G}_\theta : (x_t, t) \to x_\eta$, CMs are able to sample the r.v. $\mathbf{x}_\eta = \tilde{G}_\vartheta(\mathbf{x}_T, T)$ with $\mathbf{x}_T \sim \mathcal{N}(\mathbf{0}, \boldsymbol{I})$ in a single step, see Kim et al. (2023) for details. Two-step CMs achieve superior quality by re-noising $\mathbf{x}_\eta = \tilde{G}_\vartheta(\mathbf{x}_T, T)$ following Equation 2 for some intermediate time $s \in (\eta, T)$ followed by $\tilde{G}_\vartheta(\mathbf{x}_s, s)$ to bring back $\mathbf{x}_s$ close to $\mathbf{x}_0$. Multi-step CMs apply this strategy recursively, in 4 to 8 steps, achieving top performance while retaining computational efficiency (Kim et al., 2024). Recent examples of few-step Bayesian image restoration methods based on CMs include, for instance, the fine-tuned method CoSIGN (Zhao et al., 2025), the zero-shot method CM4IR (Garber & Tirer, 2025), and the zero-shot method LATINO (Spagnoletti et al., 2025) which uses a so-called Stable Diffusion latent CM with semantic conditioning via text prompting.

**LoRA fine-tuning of diffusion models** Low-Rank Adaptation (LoRA) (Hu et al., 2022) is a technique designed to fine-tune large attention-based architectures such as DMs in a manner that achieves comparable performance to full fine-tuning at a fraction of the computational cost. This is achieved by adding a trainable low-rank correction $\Delta_\theta$ to the model's original attention weights, which remain frozen during fine-tuning, and exploiting structural properties of attention layers in order to optimize the weights $\theta + \Delta_\theta$ at a reduced cost. LoRA fine-tuning has been widely applied to DMs. For example, in the context of DM distillation, CM-LORA strategies distill a pre-trained DM $G_\theta(\cdot, t)$ into a CM $\tilde{G}_\vartheta(\cdot, t) = G_{\theta+\Delta_\theta}(\cdot, t)$ by adjusting $\Delta_\theta$ with a CM objective combining DSM and adversarial training (Luo et al., 2023).

**Deep unfolding** (also known as deep unrolling) is a deep learning paradigm that transforms iterative optimization algorithms with a fixed number of iterations into deep neural network architectures, whereby the algorithm's steps become trainable layers in a network that is trained end-to-end (Monga et al., 2021). This approach, also known as deep unrolling, provides a powerful template for designing interpretable network architectures that incorporate $y$ and the operator $\boldsymbol{A}$ in a manner that is modular and explicit (e.g., via proximal operator layers), together with trainable elements (e.g. U-Net modules) that can be recognized as data-driven regularization terms. Because $\boldsymbol{A}$ is provided explicitly during training, unfolded networks can be easily trained to handle a range of operators (e.g., motion blurs, as represented by a database) and then instantiated for a specific operator during inference. To date, deep unfolding has primarily been implemented by unrolling optimization algorithms. A main innovation of this paper is the unfolding of a stochastic Langevin sampling algorithm instead (specifically, the LATINO algorithm of Spagnoletti et al. (2025)). Also, to the best of our knowledge, this is the first instance of deep unfolding in the context of DMs.

**Langevin PnP methods for image restoration.** DMs can serve as highly informative PnP priors within Langevin Markov chain Monte Carlo posterior samplers. Notably, Kemajou Mbakam et al. incorporates a DM within a PnP unadjusted Langevin algorithm (ULA) (Laumont et al., 2022) to estimate the posterior mean $\mathrm{E}(\boldsymbol{x}|\boldsymbol{y})$. Similarly, Coeurdoux et al. (2024b) embeds a DM within a split-Gibbs sampler (Vono et al., 2019), which is equivalent to a noisy ULA (Pereyra et al., 2023). With regards to other stragies not based on DMs, we note Coeurdoux et al. (2024a) and Melidonis et al. (2024) which explore the use of normalizing flows as PnP priors, while Holden et al. (2022) proposes a general theoretical framework for incorporating variational autoencoder (VAE) and generative adversarial network (GAN) priors.

**LAtent consisTency INverse sOlver (LATINO)** is a state-of-the-art zero-shot posterior sampling technique derived from embedding a CM prior $\tilde{G}_\vartheta(\cdot, t)$ within an overdamped Langevin diffusion process of the form $d\mathbf{x}_{0,s} = \nabla \log p(y|\mathbf{x}_{0,s})\mathrm{d}s + \nabla \log p(\mathbf{x}_{0,s})\mathrm{d}s + \mathrm{d}\mathbf{w}_s$, which converges to the posterior of interest $p(x_0|y)$ as $s \to \infty$, see Spagnoletti et al. (2025). In its simplest form, LATINO to sample from $p(x_0|y)$ is based on the following recursion[1]: for all $k \in \mathbb{N}$

$$\tilde{\mathbf{x}}_0^{(k+1)} = \mathrm{prox}_{\delta g_y}(\mathbf{x}_0^{(k)}), \qquad \texttt{implicit gradient step, } g_y : x_0 \mapsto -\log p(y|x_0)$$

$$\mathbf{x}_{t_\delta}^{(k+1)} = \sqrt{\bar{\alpha}_{t_\delta}}\tilde{\mathbf{x}}_0^{(k+1)} + \sqrt{(1-\bar{\alpha}_{t_\delta})}\boldsymbol{\epsilon}_{k+1}, \quad \texttt{sample } \mathbf{x}_{t_\delta,k+1} \sim p(\cdot|\mathbf{x}_{0,k}) \texttt{ by using forward SDE 2}$$

$$\mathbf{x}_0^{(k+1)} = \tilde{G}_\vartheta(\mathbf{x}_{t_\delta}^{(k+1)}, t_\delta), \qquad \texttt{transport } \mathbf{x}_{t_\delta}^{(k+1)} \texttt{ back to } \mathbf{x}_0^{(k+1)} \texttt{ with ODE 4} \qquad (5)$$

where $\delta, t_\delta$ are step-sizes related to the time-discretization of the Langevin process, $\boldsymbol{\epsilon}_{k+1} \sim \mathcal{N}(\mathbf{0}, \boldsymbol{I})$, and the proximal step $\mathrm{prox}_{\delta g_y}(s) = \arg\min_{x_0 \in \mathbb{R}^d} g_y(x_0) + \frac{1}{2\delta}\|x_0 - s\|_2^2/2$ with $g_y : x_0 \mapsto -\log p(y|x_0)$ is equivalent to an implicit Euler step on $g_y$, see Spagnoletti et al. (2025) and Appendix B for details. For linear Gaussian observation models of the form $y = \boldsymbol{A}x + \mathbf{n}$ with noise covariance $\sigma^2 \boldsymbol{I}$, $\mathrm{prox}_{\delta g_y}(x) = (\delta \boldsymbol{A}^\top \boldsymbol{A} + \sigma^2 \boldsymbol{I})^{-1}(\delta \boldsymbol{A}^\top y + \sigma^2 x)$, which can be computed efficiently by using a fast linear solver or a specialised scheme. Alternatively, for challenging linear operators and non-linear problems, we use an appropriate optimizer to compute an approximate solution of $\mathrm{prox}_{\delta g_y}(s) = \arg\min_{x_0 \in \mathbb{R}^d} g_y(x_0) + \frac{1}{2\delta}\|x_0 - s\|_2^2/2$, which is essentially a regularized least squares problem. It is worth mentioning that a more general form of the LATINO recursion generalizes Equation 5 by performing the noise and de-noise step on the latent space of a (deterministic) auto-encoder pair $(\mathcal{E}, \mathcal{D})$. This allows embedding CMs trained on the latent space of $(\mathcal{E}, \mathcal{D})$, notably modern stable diffusion

---

[1]The steps of the recursion in Spagnoletti et al. (2025) are presented in a different order, first sampling using 2, then the CM step, and the proximal step last. We find that starting with the proximal step performs better when unrolling few iterations.

CMs (Yin et al., 2024). In the present work, we consider CMs trained to denoise directly on pixel space, with $\mathcal{E}$ and $\mathcal{D}$ both set to the identity in Spagnoletti et al. (2025).

Integrating the CM $\tilde{G}_\vartheta$ within a Langevin sampler allows LATINO to use the likelihood function $x_0 \mapsto p(y|x_0)$ directly and exactly - through its proximal operator - without the need for approximations[2]. This represents a significant advantage over prevalent alternative DM-based posterior sampling approaches, which attempt to embed the likelihood into the DM itself — a process that is typically intractable and reliant on likelihood approximations because of time inhomogeneity. As a result, LATINO outperforms competing zero-shot DMs in sampling accuracy, with a computational cost comparable to state-of-the-art multi-step CMs (i.e., in the order of 4-8 iterations). The proposed UD$^2$M framework unfolds LATINO within a $y$-conditional distilled DM, which is trained end-to-end in a supervised manner for posterior sampling.

## 3 Proposed framework

### 3.1 UD$^2$Ms: Diffusion model unfolding and distillation for multi-step posterior sampling

We are now ready to present our proposed methodology for learning a posterior sampler for $(\mathbf{x}_0|\mathbf{y} = y)$, through unfolding and distillation of a DM that has been trained to sample $\mathbf{x}_0$ unconditionally (i.e., from the prior $p(x_0)$). We assume the availability of a DM $G_\theta(\cdot, t)$ pre-trained to approximate $G_\theta(x_t, t) \approx \mathbb{E}(\mathbf{x}_0|\mathbf{x}_t = x_t)$ with $\mathbf{x}_t = \sqrt{\overline{\alpha}_t}\mathbf{x}_0 + \sqrt{1 - \overline{\alpha}_t}\boldsymbol{\epsilon}_t$ following Equation 2. From this DM, we seek to derive a distilled model $L_\vartheta$ that, in a single step, samples approximately from the conditional distribution $p(\mathbf{x}_0|y, x_t)$. This distilled model $L_\vartheta$ is then embedded within a multi-step sampling scheme that iteratively generates $\mathbf{x}_t^{(k)} \sim p(\mathbf{x}_t|\mathbf{x}_0 = x_0^{(k-1)})$ exactly through the noising process of Equation 2, followed by approximate sampling of $\mathbf{x}_0^{(k)} \sim p(\mathbf{x}_0|y, x_t^{(k)})$ by using $L_\vartheta(y, x_t^{(k)}, t)$. The scheme starts at $t = T$ by sampling from the marginal $\mathbf{x}_T \sim \mathcal{N}(\mathbf{0}, \boldsymbol{I})$, and the timestep $t$ is reduced across iterations to reach $t = 0$ in a few steps (e.g., $N = 3$ steps). This scheme is summarized in Algorithm 1. Importantly, unlike zero-shot methods, here we consider that the likelihood function $p(y|x_0)$ associated to $p(\mathbf{x}_0|y, x_t)$ belongs a pre-determined (parametrized) class of likelihood functions of interest that is fixed during the training of $L_\vartheta$. By training over a wide range of sampled likelihoods from the determined class, we retain flexibility to specify the exact parameters of the likelihood during inference time. Specializing $L_\vartheta$ in this manner will lead to significant accuracy and computational cost gains for likelihood models belonging to the training distribution when compared to zero-shot strategies that are fully likelihood agnostic during training. Note that $L_\vartheta$ samples the conditional random variable $(\mathbf{x}_0|y, \mathbf{x}_t)$, we do not seek to sample $(\mathbf{x}_0|y)$ directly.

Our UD$^2$M approach to construct $L_\vartheta$ combines distillation with deep-unfolding, whereby we obtain a trainable architecture from unfolding an iterative algorithm into a sequential scheme (Monga et al., 2021). As mentioned previously, a main novelty of this work is to unfold a Langevin sampler to construct a generative model, which we train end-to-end. More precisely, given $t \in [0, T]$ and $\mathbf{x}_t = x_t$, we implement $L_\vartheta(y, x_t, t)$ by unfolding $K$ iterations of the LATINO Langevin sampling algorithm of Equation 5, modified to target the conditional distribution $p(\mathbf{x}_0|y, x_t)$. The precise construction of $L_\varphi$ is illustrated in Figure 2. Within each LATINO module, the prior is represented by a distilled DM $G_{\theta+\Delta_\theta}$ derived from $G_\theta$ by LORA fine-tuning, where $\Delta_\theta$ denotes LORA adaptation weights which we will train end-to-end for conditional sampling (with $\theta$ frozen), and where the likelihood $p(y, x_t|x_0) = p(y|x_0)p(x_t|x_0)$ associated with the posterior $p(\mathbf{x}_0|y, x_t)$ is involved via its proximal operator $\text{prox}_{\delta g_{y,x_t}}(s) = \arg\min_{x_0 \in \mathbb{R}^d} g_{y,x_t}(x_0) + \frac{1}{2\delta}\|x_0 - s\|_2^2/2$ where $g_{y,x_t} = -\log p(y, x_t|x_0)$. As mentioned previously, for linear Gaussian observation models of the form $y = \boldsymbol{A}x + \mathbf{n}$ with noise variance $\sigma^2$, this operator is given for all $x \in \mathbb{R}^d$ by

$$\text{prox}_{\delta g_{y,x_t}}(x) = \Sigma^{-1}\left(\frac{\boldsymbol{A}^\top y}{\sigma^2} + \frac{x_t}{\sqrt{\overline{\alpha}_t}\sigma_t^2} + \frac{x}{\delta}\right), \tag{6}$$

---

[2]The Langevin diffusion is a time-homogeneous process and the resulting iterates $\mathbf{x}_0^{(k)}$ are asymptotically ergodic, converging to a $\delta$-neighbourhood of $p(x_0|y)$ as $k \to \infty$. The iterates are not traveling backwards in time through an inhomogeneous process and therefore there is no need to compute the likelihood of $y$ w.r.t. a noisy version of $\mathbf{x}_0$, as is the case in conditional DMs.

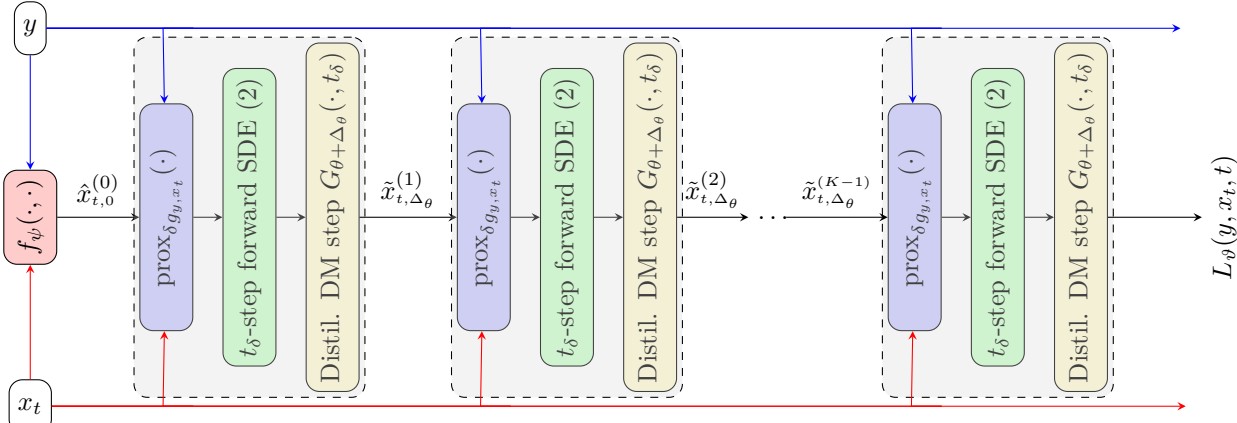

Figure 2: Diagram of the proposed conditional sampling architecture, $L_\vartheta(y, x_t, t)$ derived by deep unfolding $K$ LATINO iterations (Spagnoletti et al., 2025). The prior is introduced via a pre-trained unconditional DM $G_\theta$ with LoRA adaptation $\Delta_\theta$, while the observation model, measurement $y$ and noisy state $x_t$ are involved via $g_{y,x_t} = -\log p(y, x_t|x_0)$. The first module is initialized by an estimator $f_\psi$ such as RAM (Terris et al., 2025), or by setting $f_\psi(x_t, y) = A^\dagger y$. The unfolded network is finetuned and distilled to sample from $(\mathbf{x}_0|y, x_t)$.

where $\sigma_t^2 = (1 - \overline{\alpha}_t)/\overline{\alpha}_t$ and the matrix $\Sigma$ is defined as follows

$$\Sigma = \left( \frac{A^\top A}{\sigma^2} + \frac{I}{\sigma_t^2} + \frac{I}{\delta} \right) \tag{7}$$

For other models, we solve $\text{prox}_{\delta g_{y,x_t}}(s) = \arg\min_{x_0 \in \mathbb{R}^d} g_{y,x_t}(x_0) + \frac{1}{2\delta}\|x_0 - s\|_2^2/2$ with $g_{y,x_t} = -\log p(y, x_t|x_0)$ approximately by using Adam (Kingma & Ba, 2014) with $x_0$ as warm-starting.

Because of the low-rank structure of $\Delta_\theta$, and because we used tied weights across LATINO modules, the number of trainable parameters is usually of order of 0.1% of the number contained in $\theta$, which makes adversarial training significantly more stable. Operating with a reduced number of parameters also greatly simplifies learning and storing adaptations of $G_\theta$ tailored for specific classes of problems of interest, and allows us to easily instantiate problem-specific variants of $L_\vartheta$ during inference. Moreover, Equation 6 can be computed cheaply when a singular value decomposition of $A$ is available. Otherwise, if computing an exact solution efficiently is not possible, we use a specialised sub-iteration, a randomized linear solver, or some steps of a convex optimization scheme (Chambolle & Pock, 2016).

Lastly, to keep the number of LATINO modules $K$ small without compromising accuracy, we warm-start the first LATINO module by using auxiliary neural estimator $\hat{x}_0 = f_\psi(\mathbf{y}, \mathbf{x}_t)$ as input module, parametrized by a set of weights $\psi$. For example, in our experiments we use the recently proposed foundational model RAM (Terris et al., 2025), with stacked operators $(A, I)$ to take $(y, x_t)$ as input. Alternatively, one can also initialize with $A^\dagger y$. Accordingly, the set of trainable parameters of $L_\vartheta$ is $\vartheta = (\Delta_\theta, \psi)$ if $f_\psi$ is finetuned for additional performance; otherwise we have $\vartheta = \Delta_\theta$ if $\psi$ is frozen or if the first LATINO module is initialized directly with $A^\dagger y$. The proposed architecture is summarized in Figure 2 below.

Before discussing our strategy for training $L_\vartheta$, it is worth noting that one could in principle design $L_\vartheta$ to target the posterior $p(\mathbf{x}_0|y)$ directly, by using unfolding and distillation but bypassing fully the multi-step scheme. However, one-step samplers are notoriously harder to train, and the resulting samples are often of inferior quality (Luhman & Luhman, 2021; Meng et al., 2023; Song et al., 2023c; Salimans & Ho, 2022). Also, sampling $p(\mathbf{x}_0|y)$ directly with LATINO would require unfolding a larger number of iterations compared to sampling $p(\mathbf{x}_0|y, x_t)$, hence offsetting the computational benefits of targeting $p(\mathbf{x}_0|y)$ directly. This is due to the fact that the additional information provided by $\mathbf{x}_t = x_t$ leads to a posterior distribution that is much better conditioned and thus easier to sample by Langevin sampling. By using multi-step sampling, we obtain excellent sample quality with a reduced number of NFEs (e.g., we use $K = 3$ and $N = 3$ steps).

---

**Algorithm 1** Conditional Diffusion Sampling

---

**Require:** Observation $y$, Time-grid $0 = t_0 < t_1 < \cdots < t_N = T$,

1: Sample $x_{t_N} \sim \mathcal{N}(0, \boldsymbol{I})$          ▷ Initialize reversed diffusion

2: **for** $n = N, \ldots, 1$ **do**

3:      Set $\hat{x}_0 \leftarrow \tilde{x}_{t_n, K}^{\Delta_\theta}(x_{t_n}, y)$ using $L_\vartheta$ (see Figure 2)      ▷ Unfolded sample targeting $p_0(x_0 \,|\, x_{t_n}, y)$

4:      Sample $x_{t_{n-1}} \sim p_{t_{n-1}}(x_{t_{n-1}} \,|\, \hat{x}_0, x_{t_n})$          ▷ Reverse DDIM step

5: **end for**

6: **return** $x_{t_0}$

---

## 3.2 Training objective

To train our network to sample the conditional random variable $(\mathbf{x}_0|y, \mathbf{x}_t)$, we take inspiration from Kim et al. (2024); Bendel et al. (2023) and combine reconstruction and adversarial losses, which has been shown to greatly improve for sample generation quality as opposed to using a reconstruction loss alone. More precisely, we adapt the objective of Kim et al. (2024) and use the following objective based on an additional adversarial loss with jointly-optimized weights $\phi$

$$\underset{\Delta_\theta, \psi}{\arg\min} \quad \mathcal{L}^G(\Delta_\theta, \psi) \triangleq \mathcal{L}_{\mathrm{Adv}}(\Delta_\theta, \psi, \phi) + \omega_{\ell_2} \mathcal{L}_{\ell_2}(\Delta_\theta, \psi) + \omega_{\mathrm{PS}} \mathcal{L}_{\mathrm{PS}}(\Delta_\theta, \psi), \tag{8}$$

$$\underset{\phi}{\arg\max} \quad \mathcal{L}^D(\phi) \triangleq \mathcal{L}_{\mathrm{Adv}}(\Delta_\theta, \vartheta, \phi) + \omega_{\mathrm{GS}} \mathcal{L}_{\mathrm{GS}}(\phi), \tag{9}$$

where $\omega_{\ell_2}, \omega_{\mathrm{PS}}, \omega_{\mathrm{GS}} > 0$ are regularization hyper-parameters and the remaining loss terms are defined as follows, where we let $t$ be uniformly distributed on $[1, T]$. First, $\mathcal{L}_{\ell_2}(\Delta_\theta, \vartheta)$ is the $\ell_2^2$-regression loss commonly encountered in DSM on the pixel space,

$$\mathcal{L}_{\ell_2}(\Delta_\theta, \psi) = \mathbb{E}_{t, \mathbf{x}_t, \mathbf{y}, \mathbf{x}_0} \left[ ||\mathbf{x}_0 - L_{\Delta_\theta, \psi}(\mathbf{x}_t, \mathbf{y})||_2^2 \right]. \tag{10}$$

Second, $\mathcal{L}_{\mathrm{PS}}(\Delta_\theta, \vartheta)$ promotes perceptual quality via the Learned Perceptual Image Patch Similarity (LPIPS) loss (Zhang et al., 2018)

$$\mathcal{L}_{\mathrm{PS}}(\Delta_\theta, \psi) = \mathbb{E}_{t, \mathbf{x}_t, \mathbf{y}, \mathbf{x}_0} \left[ \mathrm{LPIPS}(\mathbf{x}_0, L_{\Delta_\theta, \psi}(\mathbf{x}_t, \mathbf{y})) \right]. \tag{11}$$

Third, the adversarial loss, $\mathcal{L}_{\mathrm{Adv}}$, is based on Goodfellow et al. (2014)

$$\mathcal{L}_{\mathrm{Adv}}(\Delta_\theta, \vartheta, \phi) = \mathbb{E}_{\mathbf{x}_0, \mathbf{y}} \left[ \log \left( \varsigma(D_\phi(\mathbf{x}_0; \mathbf{y})) \right) \right] + \mathbb{E}_{t, \mathbf{x}_t, \mathbf{y}} \left[ \log \left( 1 - \varsigma(D_\phi(L_{\Delta_\theta, \psi}(\mathbf{x}_t, \mathbf{y}), \mathbf{y})) \right) \right] \tag{12}$$

where $\varsigma : \mathbb{R} \to \mathbb{R}$ is the sigmoid function. The discrimination $D_\phi$ is a CNN adapted from Radford et al. (2016); Bendel et al. (2023), which we train to maximize the discriminator loss $\mathcal{L}^D(\phi)$. To regularize the discriminator, we follow Mescheder et al. (2018) and introduce a gradient-sparsity regularizer designed to promote Lipschitz regularity of $D_\phi$, defined by

$$\mathcal{L}_{\mathrm{GS}}(\phi) = \mathbb{E}_{t, \mathbf{x}_t, \mathbf{y}, \mathbf{x}_0, \mathbf{u}} \left[ \left\| \nabla_x D_\phi \left( \mathbf{u}\mathbf{x} + (1 - \mathbf{u}) L_{\Delta_\theta, \psi}(\mathbf{x}_t, \mathbf{y}), \mathbf{y} \right) \right\|^2 \right], \tag{13}$$

where $\mathbf{u} \sim \mathcal{U}(0, 1)$ is independent of $\mathbf{x}_0$ and $\mathbf{y}$. Note that our presentation of the training objective assumes RAM initialization, so the trainable parameters of $L_\vartheta$ are $\vartheta = (\Delta_\theta, \psi)$. If $\psi$ is frozen or if the first LATINO module is initialized directly with $\boldsymbol{A}^\dagger y$ and $x_t$, then we use the same training objective reduced to $\vartheta = \Delta_\theta$.

As mentioned previously, using Equation 8 is closely related to training $L_{\Delta_\theta, \psi}$ as a consistency trajectory model (Kim et al., 2024) to sample from $p(x|y, x_t)$. Similarly to CMs, our distilled model is trained for sampling, not for denoising. However, CMs are deterministic whereas $L_{\Delta_\theta, \psi}$ is a stochastic map, and we do not seek to enforce consistency of paths as a result. Also, we use a discriminator $D_\phi$ that conditions on $y$ only, whereas training a CM for $(\mathbf{x}_0|y, \mathbf{x}_t)$ by following Kim et al. (2024) would require using a discriminator that conditions on both $y$ and $x_t$. This would also require a larger and more complex discriminator architecture

with a time embedding, which we do not consider here, as we find that a lighter discriminator that conditions on $y$ only suffices for correctly training the proposed UD$^2$M sampler.

Before concluding this section, we present two alternative interpretations of the proposed deep unfolded architecture. Both are based on deep unfolding of discretizations of the same overdamped Langevin process that underlies LATINO, specifically the PnP-ULA scheme introduced by Laumont et al. (2022). In the first interpretation, our architecture is viewed as a deep unfolding of PnP-ULA using a pre-trained SNORE denoiser (Renaud et al., 2024) adapted via LoRA. In the second, it is interpreted as a deep unfolding of the DM-within-ULA scheme proposed by Mbakam et al. (2024). In both cases, the likelihood is incorporated through its proximal operator (Durmus et al., 2018), rather than through the more conventional gradient-based approach. Among these interpretations, we find the deep unfolding of LATINO to be the most meaningful. LATINO is itself a distilled few-step scheme that incorporates the likelihood through its proximal operator and generalizes naturally to image-text latent DMs such as Stable Diffusion (Yin et al., 2024), which are seeing increasing adoption. Developing UD$^2$M samplers by finetuning latent DMs is a natural extension of this work and main direction for future research.

## 4 Experiments

### 4.1 Experimental setup.

**Preliminaries.** To demonstrate the effectiveness of the proposed method, we now report a series numerical experiments and comparisons with competing approaches from the state of the art. In our experiments, we consider three forms of image deblurring (Gaussian, uniform and motion deblurring), two forms of image inpainting (box and uniformly randomly missing pixels), image super-resolution (SR) by a factor 4, as well as the removal of artifacts from aggressive JPEG compression (as an example of a non-linear restoration problem). Moreover, we assess the performance of our method qualitatively and quantitatively by computing the following quantitative metrics: the Peak Signal-to-Noise Ratio (PSNR) as distortion metric (Wang et al., 2004), the Learned Perceptual Image Patch Similarity (LPIPS) for perceptual quality (Zhang et al., 2018), and the Frechet Inception Distance (FID) with inception-v3 embeddings for sampling accuracy (Heusel et al., 2017).

**Datasets and experimental conditions.** We use the following two public datasets in our experiments - ImageNet (Russakovsky et al., 2015) and LSUN Bedroom (Yu et al., 2015) - which have been used extensively in prior work related to image restoration with DMs and CMs in particular. For both datasets, we crop images to a size of $256 \times 256$ pixels and normalize their pixel amplitude to the range of $[0, 1]$. For our experiments with the ImageNet dataset, we used one million images from the training set for model training, whereas for the LSUN Bedroom dataset, we used 1.2 million images for training. For computing performance metrics, we use a test set of 1500 images from ImageNet and a test set of 300 images from LSUN bedroom. Regarding the parameters of each experiment, for ImageNet restoration tasks we follow Chung et al. (2023), with the exception of motion deblurring where we follow Zhu et al. (2023). For LSUN bedroom restoration tasks, for random inpainting, Gaussian deblurring and SR ($\times 4$), we use the setup as Garber & Tirer (2025) with a noise-level $\sigma = 0.025$; we follow Zhao et al. (2025) for box inpainting. For more details about the considered experiments, please see Appendix C.3.

**Comparisons with the state of the art** We compare our proposed method with an extensive selection of zero-shot and fine-tuned diffusion models from the state of the art. For the experiments with ImageNet, we compare with the diffusion bridges I$^2$SB (Liu et al., 2023a) and CDDB (Chung et al., 2023), which are trained specifically for posterior sampling for ImageNet and for particular tasks. In addition, we compare with the two zero-shot methods DiffPIR (Zhu et al., 2023) and DDRM (Kawar et al., 2022), which rely on the same pretrained DM for ImageNet as our method, but differ in how they incorporate the data fidelity term.

For the experiments with LSUN bedroom, we use the same methods mentioned previously and two additional ones that rely on a CM trained and distilled for this dataset: the zero-shot method CM4IR (Garber & Tirer, 2025), and CoSIGN (Zhao et al., 2025), a learning-based method which upgrades an unconditional CM into a conditional CM for posterior sampling by learning an operator-specific ControlNet guidance that is

attached onto the CM (Zhao et al., 2025). Our experiments show that CoSIGN's ControlNet guidance is more computationally efficient than our unfolding approach, but lacks flexibility and leads to less accurate results as it does not use an explicit likelihood function.

**Implementation of the proposed method** We implement our method as follows. For our deep unfolded architecture, we use $K = 3$ LATINO modules with a pre-trained diffusion model denoiser[3]; we use the U-NET architecture of Ho et al. (2020) with time-embeddings and attention layers trained on a linear noise schedule $\beta_1 = 10^{-4}$ up to $\beta_{1000} = 0.02$. With regards to training, we use the training objective detailed in Section 3.2 and LoRA adaptation with rank 5, while keeping original weights $\theta$ frozen. Moreover, for each experiment, with the exception of JPEG artifact removal, we implement and train two versions of our model. In one version, the first LATINO module is initialized directly with a weighted average of the inputs $A^\dagger y$ and $x_t$. In the other, the first LATINO module is initialized by a RAM module (Terris et al., 2025) with stacked operators $(\boldsymbol{I}, \boldsymbol{A})$ that take $x_t$ and $y$ as input, and which we train end-to-end together with he LoRA adaptation $\Delta_\theta$ by using the objective of Section 3.2. We do not consider RAM initialization for the JPEG artifact removal task because RAM is not suitable for non-linear image restoration problems. For more details about the model's architecture, training parameters, GPU and time consumption, please see Appendix C.

**Extension to non-linear inverse problems** Similarly to Spagnoletti et al. (2025), to apply our method to JPEG artifact restoration and other non-linear image restoration problems we solve $\mathrm{prox}_{\delta g_y}(s) = \arg\min_{x_0 \in \mathbb{R}^d} g_y(x_0) + \frac{1}{2\delta}\|x_0 - s\|_2^2/2$ with $g_{y,x_t} = -\log p(y, x_t|x_0)$ approximately by using an optimizer -we use Adam (Kingma & Ba, 2014) with $s$ as warm-starting. We do not use RAM initialization.

## 4.2 Results

Table 1 and Table 2 below summarize the performance metrics for our proposed method, with (w/) and without (wo) RAM initialization, for the considered ImageNet and LSUN Bedroom experiments respectively. For comparison, we also report the performance metrics for the alternative approaches from the state of the art. The results for the ImageNet motion deblurring task are reported separately in Table 7, as there are no publicly available finetuned models for posterior sampling for that task for comparison. As mentioned previously, we do not use RAM initialization for JPEG artifact removal because RAM does not support non-linear problems, and we do not use RAM initialization for box inpainting because it is a lightweight model with a receptive field that is not sufficiently large to meaningfully inpaint large regions.

We observe from Table 1 and Table 2 that the proposed approach performs very strongly relative to the state of the art across all tasks and all metrics, especially when RAM initialization is used. Notably, our approach very clearly outperforms the alternative methods from the state-of-the-art in terms of FID and LPIPS scores, indicating excellent perceptual and sampling quality. For example, for the SR ($\times 4$) task, our method with RAM initialization achieves a FID score of 11.9 for ImageNet and 19.48 for LSUN Bedrooms, largely outperforming the second best methods (CDDB and CoSIGN respectively) which achieve scores of 19.88 and 40.48. In addition, the proposed approach is highly computationally efficient, as it only requires in the order of 10 NFEs per posterior sample, a key computational advantage relative to non-distilled methods. This illustrates the benefit of training with the CM objective of Equation 8 that induces distillation. Having said that, out method is not as efficient as CoSIGN, which does not have the cost of evaluating $K = 3$ unfolded LATINO modules per step of the multi-step scheme.

For illustration and qualitative evaluation, Figure 3, Figure 4, and Figure 5 depict examples of posterior samples for the ImageNet SR ($\times 4$), Gaussian deblurring and JPEG restoration tasks, respectively. We observe that our proposed approach (w/ RAM initialization) delivers sharp posterior samples that recover a significant amount of the fine detail in the ground truth images, and without noticeable noise or color bias. With regards to the LSUN experiments, Figure 6 and Figure 7 present results for the SR ($\times 4$) and box inpainting tasks, respectively. Again, we observe that our proposed approach (w/ RAM initialization) delivers sharp posterior samples with fine detail and no noticeable noise or color bias, in agreement with the strong performance metrics reported in Table 2.

---

[3]For ImageNet, we use the un-(class)conditional model `https://github.com/openai/guided-diffusion/` and for LSUN bedroom the model `https://github.com/pesser/pytorch_diffusion/`

Table 1: Quantitative results for the deblurring and super-resolution tasks. PSNR [dB] (↑), LPIPS (↓) and FID (↓) results on ImageNet dataset with noise level 0.01. Comparisons with I²SB and CDDB are taken from Chung et al. (2023).

| Methods | NFEs | Deblurring Gaussian | | | Uniform | | | SR ×4 | | | JPEG QF=10 | | |
|---|---|---|---|---|---|---|---|---|---|---|---|---|---|
| | | PSNR | LPIPS | FID | PSNR | LPIPS | FID | PSNR | LPIPS | FID | PSNR | LPIPS | FID |
| Ours (wo RAM) | 9 | **38.77** | 0.02 | 4.61 | 35.57 | 0.02 | 11.14 | 24.42 | 0.15 | 20.69 | **27.52** | **0.18** | 35.16 |
| Ours (w/ RAM) | 12 | 35.97 | **0.01** | **3.30** | **36.96** | **0.01** | **2.69** | **26.70** | **0.08** | **11.9** | - | - | - |
| CDDB | 1000 | 37.02 | 0.06 | 5.01 | 31.26 | 0.19 | 23.15 | 26.41 | 0.2 | 19.88 | 26.34 | 0.26 | **19.48** |
| I2SB | 1000 | 36.01 | 0.07 | 5.8 | 30.75 | 0.2 | 23.01 | 25.22 | 0.26 | 24.13 | 26.12 | 0.27 | 20.35 |
| DiffPIR | 100 | 28.10 | 0.13 | 21.53 | 31.44 | 0.10 | 20.20 | 20.39 | 0.36 | 70.45 | - | - | - |
| DDRM | 20 | 36.73 | 0.07 | 4.34 | 29.21 | 0.21 | 19.97 | 26.05 | 0.27 | 46.49 | 26.33 | 0.33 | 47.02 |

Table 2: Deblurring, inpainting and super-resolution. PSNR [dB] (↑), LPIPS (↓) and FID (↓) results on LSUN bedroom dataset with noise level 0.025. Comparisons with I²SB and, CDDB and CoSIGN on the Box inpainting and SR (×4) problems are taken from Zhao et al. (2025). The remaining comparisons are obtained from Garber & Tirer (2025).

| Methods | NFEs | Inpainting Box | | | Random | | | Deblurring Gaussian | | | SR ×4 | | |
|---|---|---|---|---|---|---|---|---|---|---|---|---|---|
| | | PSNR | LPIPS | FID | PSNR | LPIPS | FID | PSNR | LPIPS | FID | PSNR | LPIPS | FID |
| Ours (wo RAM) | 9[a] | 22.22 | **0.11** | **8.45** | 22.32 | 0.19 | 43.3 | 26.73 | 0.26 | 88.97 | 24.51 | 0.19 | 32.71 |
| Ours (w/ RAM) | 12 | - | - | - | **27.75** | **0.06** | **8.61** | **29.67** | **0.08** | **13.23** | 25.48 | **0.12** | **19.48** |
| CoSIGN | 2 | 22.61 | 0.14 | 38.64 | 23.22 | 0.37 | - | 19.74 | 0.342 | - | 26.13 | 0.22 | 40.84 |
| CDDB | 1000 | **23.74** | 0.13 | 45.20 | - | - | - | - | - | - | **27.31** | 0.24 | 54.20 |
| I²SB | 1000 | 23.21 | 0.28 | 55.10 | - | - | - | - | - | - | 27.23 | 0.24 | 53.40 |
| DiffPIR | 100 | 13.42 | 0.34 | 66.79 | 23.8 | 0.36 | 97.69 | 27.48 | 0.32 | 64.81 | 23.83 | 0.45 | 101.92 |
| DDRM | 20 | 18.90 | 0.22 | 51.50 | 19.16 | 0.55 | - | 28.94 | 0.22 | - | 25.09 | 0.37 | - |
| CM4IR | 4 | 21.98 | 0.24 | 42.83 | 25.28 | 0.33 | 159.43 | 29.00 | 0.21 | 53.19 | 26.14 | 0.30 | 124.22 |

[a]We have implemented our method with $K = 3$ and $N = 3$ (so 9 NFEs) for all experiments, except for box inpainting where we have used $N = 5$ instead of $N = 3$ (so 20 NFEs), as this is a more challenging task.

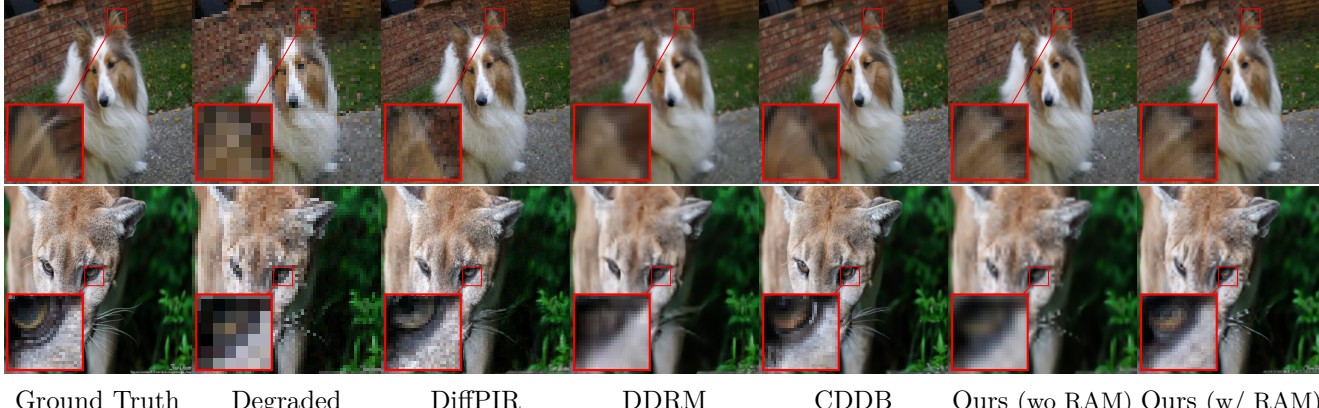

Ground Truth    Degraded    DiffPIR    DDRM    CDDB    Ours (wo RAM)   Ours (w/ RAM)

Figure 3: Comparison of posterior samples for the task SR (×4) with noise level $\sigma = 0.01$ on ImageNet 256.

## 4.3 Ablation Studies

We now proceed to examine and quantify the role of different key elements of our proposed method. This is achieved by appropriately modifying or deactivating these elements and quantifying changes in performance.

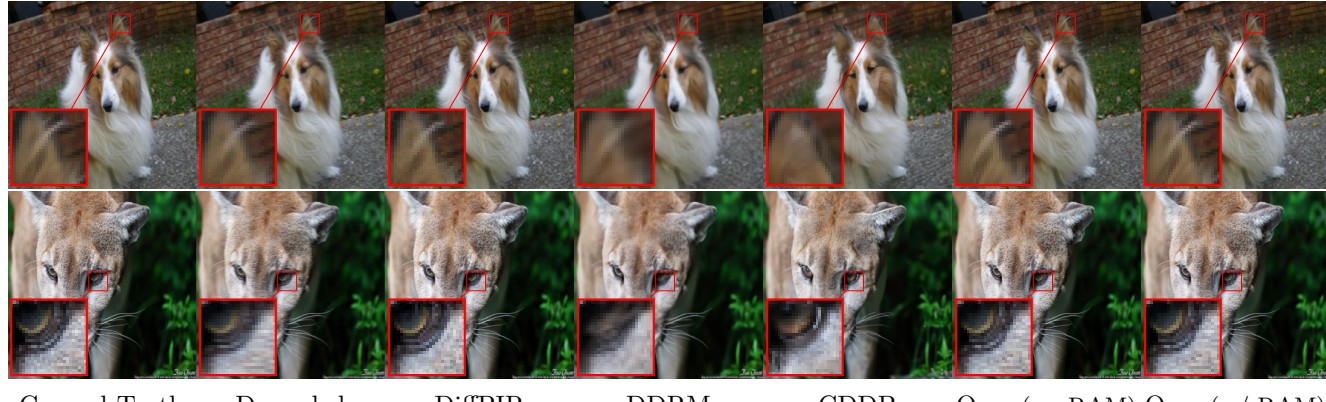

Ground Truth     Degraded     DiffPIR     DDRM     CDDB     Ours (wo RAM) Ours (w/ RAM)

Figure 4: Comparison of posterior samples for the task Gaussian deblurring with noise level $\sigma = 0.01$ on ImageNet 256.

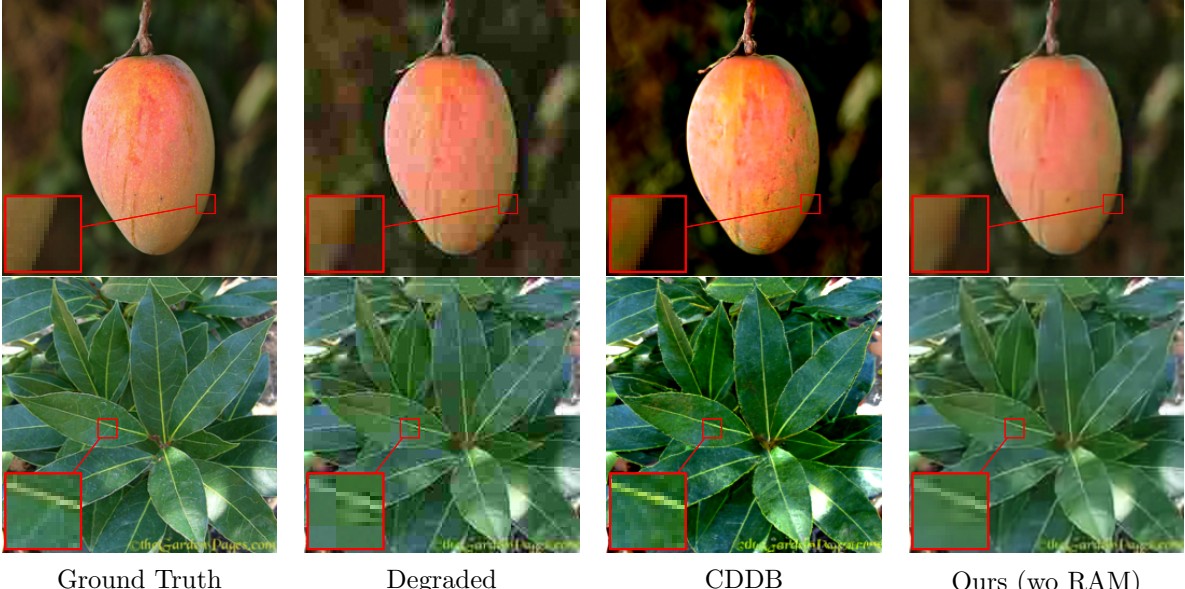

Ground Truth        Degraded        CDDB        Ours (wo RAM)

Figure 5: Comparison of posterior samples for the task JPEG artifact removal (QF=10) with noise level $\sigma = 0.01$ on ImageNet 256.

### 4.3.1 Choice of the number of sampling steps $N$

First we assess the effect of modifying the number of sampling steps $N$ in the multi-step scheme. Note that $N$ is specified during test time, so modifying it does not requires retraining the model. Table 3(a) shows the performance metrics for $N \in \{1, 3, 9, 27\}$ for the LSUN SR ($\times 4$) model with $K = 3$ LATINO modules and RAM initialization (recall that in our previous SR ($\times 4$) experiments we have used $K = 3$ and $N = 3$). We observe in Table 3(a) that the performance metrics do not change significantly with variations of $N$, indicating that the proposed approach is robust to different choices of $N$. Low values of $N$ lead to some small amount of estimation bias towards the mean, and hence to a small increase in PNSR at the expense of LPIPS and FID performance, whereas larger values of $N$ lead to a mild decrease in overall performance due to accumulation of errors (see Kim et al. (2023) for details) as well as to an increase in computational cost from additional NFEs. As mentioned previously, in our experiments we used $N = 3$, which we found to reliably

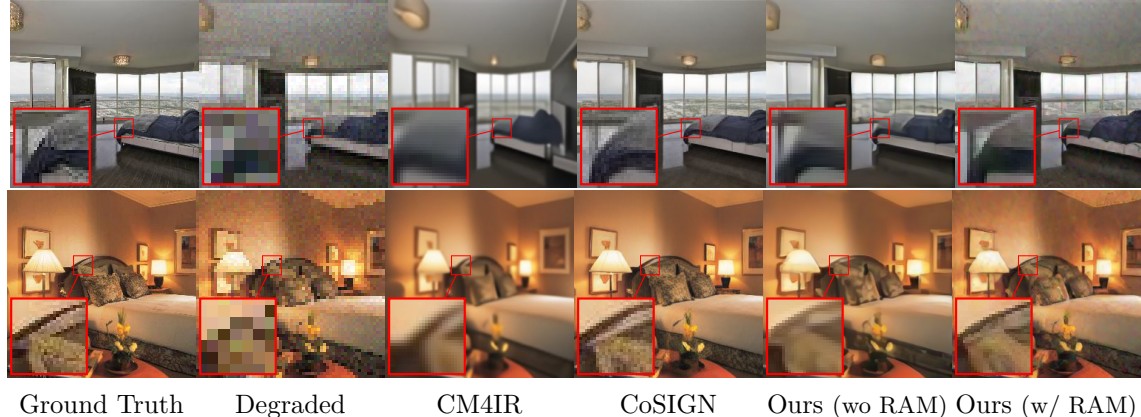

Ground Truth  Degraded  CM4IR  CoSIGN  Ours (wo RAM) Ours (w/ RAM)

Figure 6: Comparison of posterior samples for the task SR ($\times 4$) with noise level $\sigma = 0.025$ on LSUN Bedroom.

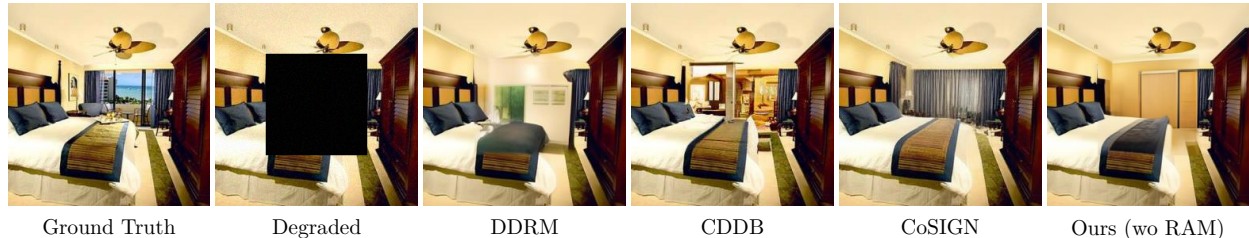

Ground Truth  Degraded  DDRM  CDDB  CoSIGN  Ours (wo RAM)

Figure 7: Comparison of posterior samples for the task box inpainting with noise level $\sigma = 0.025$ on LSUN Bedroom.

provide good performance with a low computational cost across all the considered tasks (with the exception of box inpainting, when we use $N = 5$ as the task is challenging and we don't use RAM initialization).

### 4.3.2 Effect of unfolding

We now assess the benefit of unfolding multiple LATINO modules per sampling step, as opposed to using a single LATINO module, with the same or possibly a larger number of sampling steps. Table 3(b) reports the performance metrics when retraining the network with a single LATINO module ($K = 1$) and RAM initialization. By comparing Table 3(b) with Table 3(a), we observe a noticeable deterioration in performance sampling quality as measured by FID, even if the reduction the number of LATINO modules per step is compensated by additional sampling steps. We conclude that there is a clear benefit in distilling conditional DM architectures from deep unfolding, which is consistent with the literature on deep unfolding of optimization

Table 3: Reconstruction metrics for varying number of conditional diffusion steps ($N$) and unfolded iterations ($K$).

| N | PSNR ↑ | LPIPS ↓ | FID ↓ | NFEs ↓ | N | PSNR ↑ | LPIPS ↓ | FID ↓ | NFEs ↓ |
|---|--------|---------|-------|--------|---|--------|---------|-------|--------|
| 1 | 25.93 | 0.11 | 14.07 | 4 | 1 | 25.60 | 0.11 | 17.95 | 2 |
| 3 | 25.88 | 0.10 | 13.15 | 12 | 3 | 25.61 | 0.11 | 16.59 | 6 |
| 9 | 25.54 | 0.10 | 15.10 | 36 | 9 | 24.72 | 0.12 | 20.42 | 18 |
| 27 | 25.02 | 0.11 | 17.47 | 108 | 27 | 22.82 | 0.12 | 21.93 | 54 |
| | (a) K=3 | | | | | (b) K=1 | | | |

algorithms (Monga et al., 2021). It is also worth mentioning that modifying $K$ without retraining the DM leads to severe visual artifacts (results not reported here), so it is not possible to modify $K$ during inference.

### 4.3.3 The choice of the rank of the LoRA adaptation $\Delta_\theta$

We now assess the effect of the LoRA fine-tuning rank parameter. As explained previously, rather than fine-tuning the full model, we use a LoRA approach whereby we train a low-rank correction $\Delta_\theta$ that is added to the original model weights $\theta$, which remain frozen. The number of trainable parameters in $\Delta_\theta$ is directly related to the choice of the rank of $\Delta_\theta$. Setting the rank too low deteriorates performance because it constrains the model too much, whereas setting the rank too high increases the computational cost of fine-tuning the model and can lead to some overfitting. In order to illustrate this, we train again our model for ImageNet SR ($\times 4$) by setting the rank to 2 and to 20 (in our previous results we have used rank 5). The number of trainable parameters in $\Delta_\theta$ is 165888 when the rank of $\Delta_\theta$ is 2, 414720 when the rank is 5, and 1658880 when it is 20; this represents 0.03%, 0.08% and 0.3% of the total number of parameters in $\theta$, respectively. The results for this study are summarized in Table 4. We observe that the model's performance, as measured by PSNR and LPIPS, is robust to large variations in the rank of $\Delta_\theta$. However, the rank of $\Delta_\theta$ has a noticeable effect on FID performance, which drops significantly if it is set too low, indicating a clear deterioration in sampling quality (e.g., in Table 4 the FID for rank 2 is almost twice that of rank 5).

Table 4: Ablation study evaluating the impact of varying LoRA ranks on the ImageNet dataset for the super-resolution task.

| rank($\Delta_\theta$) | 2 | 5 | 20 |
|---|---|---|---|
| PSNR | 26.05 | 26.70 | 26.32 |
| LPIPS | 0.11 | 0.08 | 0.09 |
| FID | 19.75 | 11.9 | 13.03 |
| $|\Delta_\theta|$ | 165888 | 414720 | 1658880 |

### 4.3.4 Generalization w.r.t. to the noise variance $\sigma^2$

Models obtained by deep unfolding often generalize better to different levels of measurement noise, by comparison to other end-to-end training strategies that are not aware of the value of the noise variance $\sigma^2$ during test time. To demonstrate this capability, we follow (Garber & Tirer, 2025) and evaluate our LSUN bedroom SR ($\times 4$) model trained for $\sigma = 0.025$ (with $N = 3$ and $K = 3$) in a more challenging situation ($\sigma = 0.05$). The results are summarized in Table 5. For comparison, we also report the results obtained with the CoSIGN and CM4IR models applied to the same problem (results from (Garber & Tirer, 2025, Table 2)), where CoSIGN is also finetuned for $\sigma = 0.025$ and where CM4IR, as a zero-shot method, is provided the correct value of $\sigma = 0.05$ during test time. We observe that our proposed method is able to use the value of $\sigma = 0.05$ effectively during test time, even if training was performed using $\sigma = 0.025$, similarly to zero-shot methods like CM4IR. Conversely, CoSIGN struggles to generalize to the higher noise level and exhibits poor PSNR and LPIPS metrics as a result. Again, by comparison to CM4IR, we achieve significantly better perceptual quality as reflected by a lower LPIPS metric. For completeness, Figure 8 depicts examples of observations and samples for our method when $\sigma = 0.0125$, $\sigma = 0.025$ (in-distribution), and $\sigma = 0.05$.

Table 5: Reconstruction statistics for our model applied to the SR ($\times 4$) task on LSUN bedroom, with varying noise level in the observation space.

| | Ours | | CoSIGN | | CM4IR | |
|---|---|---|---|---|---|---|
| Noise level $\sigma$ | PSNR | LPIPS | PSNR | LPIPS | PSNR | LPIPS |
| 0.025 (in-distribution) | 25.12 | 0.178 | 26.10 | 0.205 | 26.14 | 0.295 |
| 0.05 (out-of-distribution) | 24.11 | 0.214 | 20.35 | 0.509 | 25.60 | 0.320 |

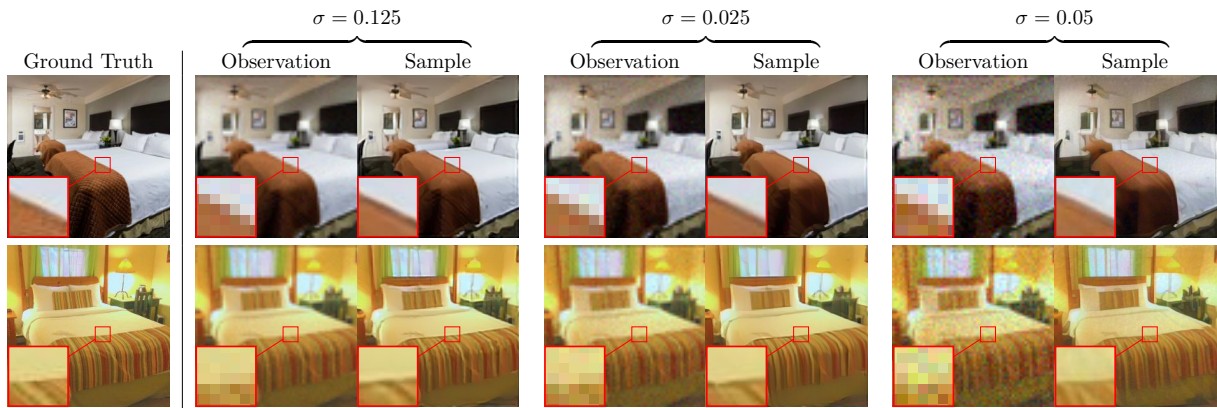

Figure 8: Comparison of posterior samples for the task SR (×4) on LSUN Bedroom, for three levels of noise.

Table 6: Reconstruction metrics computed on the LSUN Bedroom validation set using out-of-distribution forward operators for the model trained on LSUN Bedroom SR (×4).

| | CoSIGN | | Ours | |
|---|---|---|---|---|
| Task / Operator | PSNR ↑ | LPIPS ↓ | PSNR ↑ | LPIPS ↓ |
| SR (×8) - out-of-distribution | 20.11 | 0.55 | 20.40 | 0.32 |
| SR (×4) - in-distribution | 26.13 | 0.22 | 25.40 | 0.12 |

### 4.3.5 Generalization w.r.t. to more challenging degradations

We now explore the capacity of our proposed approach to deal with more challenging forward models not seen during training. For this exploratory analysis, we apply our LSUN Bedroom SR (×4) model to a more challenging SR ×8 task (we keep $\sigma = 0.025$, as in training). Again, we report comparisons with the LSUN Bedroom SR ×4 CoSIGN model. Although both models are trained for SR (×4), our method receives the correct SR (×8) forward operator during inference. Conversely, CoSIGN does not have access to the SR (×8) forward operator, as it relies on a pre-trained ControlNet guidance. Through this exploratory experiment, we seek to show that models obtained by deep unfolding are more robust to challenging degradations (or forward operators) not encountered during training, by comparison to other end-to-end training strategies. A summary of these results is presented in Table 6 and an example in Figure 9. We observe that, while both methods struggle with this difficult restoration task, our method clearly outperforms CoSIGN which is unable to use the forward operator $\boldsymbol{A}$ during test time and exhibits strong reconstruction artifacts as a result. For completeness, Figure 9 also presents the result obtained with the CM4IR zero-shot method, which also outperforms CoSIGN by using the forward operator $\boldsymbol{A}$, but suffers from some smoothing because of the approximations involved in zero-shot inference.

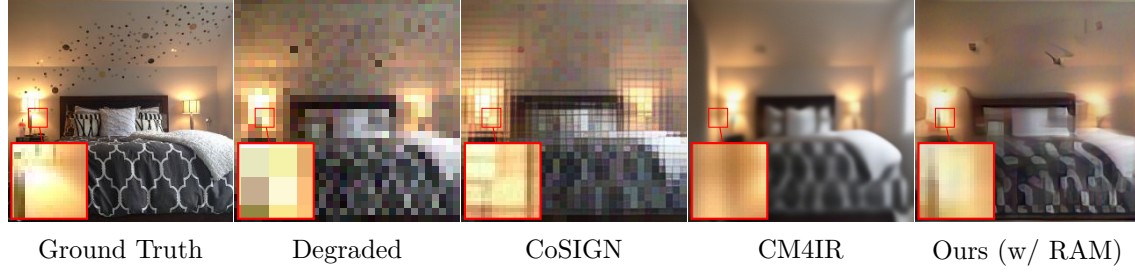

Ground Truth      Degraded      CoSIGN      CM4IR      Ours (w/ RAM)

Figure 9: Comparison of posterior samples for the task SR (×8), with noise level $\sigma = 0.025$, on LSUN Bedroom with models trained for LSUN Bedroom SR (×4), to illustrate the models' capacity to generalize to more challenging tasks.

### 4.3.6 Task specific versus universal image restoration models

In our previous experiments, we considered that $A$ is unique and known a priori, or it is not unique but belongs to a known class of operators specific for the considered task (e.g., motion blurs). In principle, one could also consider training a single common model for a much wider range of tasks. In particular, it would be interesting to train a "universal" Bayesian image restoration model, as considered in Terris et al. (2025) for MMSE estimation. Unfortunately, developing such general models requires using a much more sophisticated architecture than the one we consider here. For example, Terris et al. (2025) combines unfolding with several additional key elements not considered here (e.g., a multiscale decomposition of the operators, Krylov updates). Without these substantial improvements, our proposed approach suffers from a drop in performance when we try to generalize too widely. To illustrate this phenomenon, we trained a single posterior sampling model for the ImageNet dataset, simultaneously on Gaussian, uniform and motion deblurring, random inpainting and super-resolution tasks. The corresponding results are summarized in Table 7, where we also report the results obtained with our task specific models for comparison. We observe that the task specific models outperform the model trained jointly on all the considered tasks, and in all the considered performance metrics (the difference is most noticeable for FID performance). The extension of the UD$^2$Ms framework to support the development of general purpose posterior samplers that can tackle a wide range of image restoration tasks and a wide range of datasets, without retraining or compromising performance, is a main perspective for future work.

Table 7: Qualitative results of generalized models evaluated on the ImageNet dataset. The reported metrics include the FID score, average PSNR, and average LPIPS.

|  | Gaussian Deb | | | Motion Deb | | Uniform Deb | | | Inp 70% | | SR ($\times$4) | | |
| --- | --- | --- | --- | --- | --- | --- | --- | --- | --- | --- | --- | --- | --- |
|  | Spec | Univ | DiffPIR | Spec | Univ | Spec | Univ | DiffPIR | Spec | Univ | Spec | Univ | DiffPIR |
| PSNR | 35.97 | 34.80 | 28.1 | 33.40 | 31.10 | 36.96 | 31.35 | 31.44 | 31.15 | 30.16 | 26.70 | 25.74 | 20.39 |
| LPIPS | 0.01 | 0.03 | 0.13 | 0.04 | 0.04 | 0.01 | 0.08 | 0.1 | 0.03 | 0.06 | 0.08 | 0.12 | 0.36 |
| FID | 3.30 | 4.94 | 21.53 | 4.08 | 6.56 | 2.69 | 19.6 | 20.2 | 7.87 | 13.21 | 11.9 | 21.34 | 70.45 |

## 5 Conclusion

### 5.1 Concluding remarks and perspectives for future work

Diffusion models are powerful image priors for Bayesian computational imaging (Daras et al., 2024). The literature offers two main strategies to construct Bayesian imaging techniques based on diffusion models: Plug & Play methods that are zero-shot and hence highly flexible but reliant on approximations, and $y$-conditional diffusion models that achieve superior accuracy and speed through supervised specialization for a particular task. The framework introduced in this paper, unfolded and distilled diffusion models (UD$^2$Ms), leverages deep unfolding and distillation in order to upgrade a diffusion model representing the image prior into a few-step consistency model for posterior sampling. UD$^2$M samplers offer excellent accuracy and high computational efficiency, while retaining the flexibility required to generalize to variations in the forward operator and noise distribution during inference. A main novelty of UD$^2$M is to unfold a Markov chain Monte Carlo algorithm, namely the recently proposed LATINO Langevin sampler (Spagnoletti et al., 2025), the first example of deep unfolding of a Monte Carlo scheme.

With regards to limitations of our proposed framework and perspectives for future work. We are cognizant that diffusion models can reproduce biases stemming from unbalanced or biased training data and distribution shift, diminishing the reliability of our framework. Future work should develop appropriate guardrails so that DM-based posterior samplers can be deployed safely and reliably. Also, we hope and anticipate that these issues will be partially mitigated by progress and democratization of deep generative modeling technology and the advancement of self-supervised deep generative models. Moreover, while our samplers are more computationally efficient than some alternatives, they still rely on very large models and therefore consume a significant amount of energy and compute resources. The development of more frugal architectures and

sampling strategies is an important perspective for future work. Furthermore, as mentioned previously, it would be interesting to study architectures and training strategies to develop UD$^2$M samplers that can be applied to a wide range of image restoration tasks and datasets without the need for additional training, or in a few-shot manner, as is the case of the foundational image reconstruction model of Terris et al. (2025). In addition, as mentioned previously, many state-of-the-art DMs leverage the latent space representation of an autoencoder. For example, Stable Diffusion models (Yin et al., 2024) that allow semantic conditioning through text prompting. Extending UD$^2$M samplers to these DMs is a promising direction for future work.

Lastly, in parallel with this work, the concurrent works Wang et al. (2025); Elata et al. (2025) also seek to bridge supervised and zero-shot strategies for Bayesian imaging with diffusion models. Wang et al. (2025) relies on unfolding a conventional proximal gradient optimization scheme and uses an $\ell_2$ loss in order to obtain a conditional diffusion model, without distillation. Elata et al. (2025) proposes a novel approach to incorporate the forward operator $\boldsymbol{A}$ and the measurement $y$ directly within Transformer or UNet architectures commonly used for diffusion models. Contrasting and combining UD$^2$M with these alternative strategies is an important perspective for future work.

### 5.1.1 Broader impact statement

Computational imaging research generally poses some ethical dilemmas, especially when it involves significant methodological innovation. Although the innovations studied in the project do not raise any direct concerns, we are aware that the proposed Bayesian imaging framework could be adapted and transferred towards military applications by a hostile actor and for violating citizen privacy, for example. In addition, our framework relies on deep generative modeling technology that currently suffers from biases, which could lead to leading to unfair or discriminatory outcomes. We hope that these critical issues will be resolved by progress and democratization of generative modeling technology.

Moreover, the innovations presented in this paper fall within the "limited risk" level of the EU AI Act. We have followed the Act's recommendations for this level; i.e., a focus on the full transparency of the training data and the methods used. This also links in with our open research philosophy - we will release publicly available open-source implementations for research codes, with clear documentation and demonstrations. We see this as an essential requirement for conducting research in a democratic knowledge society.

### Acknowledgments

We are grateful to Andres Almansa, Abdul-Lateef Haji-Ali, Alessio Spagnoletti and Konstantinos Zygalakis for useful discussion. This work was supported by UKRI Engineering and Physical Sciences Research Council (EPSRC) (EP/V006134/1, EP/Z534481/1). We acknowledge the use of the HWU high-performance computing facility (DMOG) and associated support services in the completion of this work.

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

## A Solving the proximal step in Figure 2

This section outlined the approach for solving the proximal step in Figure 2 tailored to each image restoration task in our study.

**Deblurring** The degradation model for deblurring is generally express as $y = \boldsymbol{A}x + \boldsymbol{n}$, where $y$ is the observation, $\boldsymbol{n}$ is zero mean Gaussian noise with variance $\sigma^2$. The forward operator $\boldsymbol{A}$ represents a circular block matrix, and can be transformed as $\boldsymbol{A} = \mathcal{F}^\dagger \Sigma_{\boldsymbol{A}} \mathcal{F}$, where $\mathcal{F}$ and $\mathcal{F}^\dagger$ denote the fast Fourier transform (FFT) and its inverse, respectively, while $\Sigma_{\boldsymbol{A}}$ is a diagonal matrix containing the Fourier coefficients of the blurring kernel. A fast solution of the proximal step in Figure 2 is given by

$$\hat{x}_{t,k} = \mathcal{F}^\dagger \left( \left[ \frac{\Sigma_{\boldsymbol{A}}^\top \mathcal{F}y}{\sigma_y^2} + \frac{\mathcal{F}x_t}{\sqrt{\bar{\alpha}_t}\sigma_t^2} + \frac{\mathcal{F}\tilde{x}_{t,k}^{\Delta_\theta}}{\delta} \right] \Big/ \left[ \frac{\Sigma_{\boldsymbol{A}}^2}{\sigma^2} + \frac{\boldsymbol{I}}{\sigma_t^2} + \frac{1}{\delta} \right] \right). \tag{14}$$

**Inpainting** For inpainting task, the degradation process is formulated as $y = \boldsymbol{A} \circ x + \boldsymbol{n}$, where $\boldsymbol{A}$ denotes the mask operator and $\circ$ represents the elementwise multiplication operator. By using the well known Sherman–Morrison formula (Maponi, 2007), we obtain the following fast solution for the proximal step in Figure 2

$$\hat{x}_{t,k} = \kappa_k \left( \boldsymbol{I} - \kappa_k \frac{\boldsymbol{A}}{\sigma^3 + \kappa_k} \right) \circ \left( \frac{\boldsymbol{A}^\top y}{\sigma^2} + \frac{\tilde{x}_{t,k}^{\Delta_\theta}}{\delta} + \frac{x_t}{\sqrt{\bar{\alpha}_t}\sigma_t^2} \right), \tag{15}$$

where $\kappa_k = \frac{\delta\sigma_t^2}{\delta + \sigma_t^2}$ and $\circ$ denotes the elementwise multiplication operator.

**Super-resolution** In super-resolution task, we formulate the problem by assuming that the low-resolution image is a blurred, downsampled, and noisy version of the high-resolution image. Mathematically, the model is given by $y = \boldsymbol{S}_s \boldsymbol{A}x + \boldsymbol{n}$, where $\boldsymbol{S}_s$ represents the downsampler operator with factor $s$ and $\boldsymbol{A}$ represents the circular blur operator. According to Zhao et al. (2016); Zhang et al. (2021) a fast solution for the proximal step in Figure 2 is given by

$$\hat{x}_{t,k} = \mathcal{F}^\dagger \left( \frac{1}{\kappa_k} \left( \boldsymbol{r} - (\Sigma_{\boldsymbol{A}}^\top) \downarrow_s \circ_s \left( \frac{\Sigma_{\boldsymbol{A}}\boldsymbol{r}) \downarrow_s}{(\Sigma_{\boldsymbol{A}}^2) \downarrow_s + \kappa_k} \right) \right) \right), \tag{16}$$

where $\boldsymbol{r} = \Sigma_{\boldsymbol{A}}^\top \boldsymbol{S}^\top y + \kappa_k \left( \delta x_t + \sqrt{\bar{\alpha}_t}\sigma_t^2 \tilde{x}_{t,k}^{\Delta_\theta} \right)$, $\kappa_k = \frac{\sigma^2}{\bar{\alpha}_t \sigma_t^2 \delta}$ and $\circ_s$ represents distinct block processing operator with element-wise multiplication. The operator $\downarrow_s$ denotes distinct block downsampler. For more details, see Zhao et al. (2016); Zhang et al. (2021).

**Extension to non-linear tasks: JPEG** For the non-linear JPEG restoration task, the observation model is given by $y = \mathcal{A}(x) + \boldsymbol{n}$, where $\mathcal{A}$ denotes the non-linear compression operator and $\boldsymbol{n}$ is the zero-mean Gaussian noise with variance $\sigma^2$. In this case, computing the exact solution of the proximal step $\text{prox}_{\delta g_y}(s) = \arg\min_{x_0 \in \mathbb{R}^d} g_y(x_0) + \frac{1}{2\delta}\|x_0 - s\|_2^2/2$ with $g_{y,x_t} = -\log p(y, x_t|x_0)$ is not feasible. To address this challenge, we approximate the solution by using Adam optimizer (Kingma & Ba, 2014) with $s$ as warm-starting and we do not use RAM initialization.

## B Relation to Langevin Dynamics

In this appendix, we present a derivation of the LATINO kernel equation 5 as a small-timestep discretisation with a score-based prior of the regularized Langevin SDE:

$$d\mathbf{x}_{0,s} = \nabla \log p(y|\mathbf{x}_{0,s})ds + \nabla \log p(\mathbf{x}_{0,s})ds + d\mathbf{w}_s,$$

which is ergodic with respect to the posterior distribution $p(\mathbf{x}|y)$. An alternative derivation, based on a large timestep approximation of the Langevin diffusion with a consistency model prior can be found in Spagnoletti

et al. (2025). Following Pereyra et al. (2023), we consider an auxiliary variable $\mathbf{z}$ such that, conditioned on $\mathbf{z} = z$, we have $\mathbf{x} \sim \mathcal{N}(z, \delta)$. The marginal process $\mathbf{z}_{0,s} \sim (\mathbf{z}|\mathbf{x} = \mathbb{E}[\mathbf{x}_{0,s}|\mathbf{y}, \mathbf{z}_{0,s}])$ then follows the diffusion

$$
\begin{aligned}
\mathrm{d}\mathbf{z}_{0,s} &= -\frac{\|\overline{\mathbf{x}_{0,s}} - \mathbf{z}_{0,s}\|^2}{2\delta}\mathrm{d}s + \nabla \log p(\mathbf{z}_{0,s})\mathrm{d}s + \mathrm{d}\mathbf{w}_s, \\
\overline{\mathbf{x}_{0,s}} &= \mathbb{E}[\mathbf{x}_{0,s}|\mathbf{y}, \mathbf{z}_{0,s}],
\end{aligned}
\tag{17}
$$

where we express the augmented likelihood $\nabla \log p(\mathbf{y}|\mathbf{z}, \delta)$ in terms of $\overline{\mathbf{x}_{0,s}}$ and $\mathbf{z}_{0,s}$ through Fisher's Identity as in Pereyra et al. (2023). The short-time continuous dynamics of $\mathbf{z}_{0,h}$ for $h \ll 1$ can be estimated through the proximal splitting scheme

$$
\begin{aligned}
\mathbf{z}_{0,h} &\approx \mathrm{Prox}_{-h \log p}\left(\mathbf{z}_{0,h}\left(1 - \frac{h}{\delta}\right) + \frac{h}{\delta}\overline{\mathbf{x}_{0,h}} + \sqrt{2h}\epsilon\right) \\
\epsilon &\sim \mathcal{N}(0, I).
\end{aligned}
$$

Setting $h = \delta$ and approximating the proximal operator by a minimum mean-square denoiser $D_\delta$ for the convolution of $p(z)$ with $\mathcal{N}(0, \delta)$ additive noise as in Laumont et al. (2022), we obtain the stochastic noise-and-denoise step

$$
\mathbf{z}_{0,h} \approx D_\delta(\overline{\mathbf{x}_{0,h}} + \sqrt{2\delta}\epsilon).
$$

Ergodic convergence guarantees for plug-and-play discretisation of the Langevin diffusion can be found in Laumont et al. (2022). In a related work, Martin et al. (2025 (to appear) consider the use of an identical noise-and-denoise within stochastic gradient descent algorithms with provable convergence guarantees to a maximum-a-posteriori estimate. Setting $\delta = \sqrt{(1 - \bar{\alpha}_{t_\delta})/2\bar{\alpha}_{t_\delta}}$ and multiplying by a factor of $\sqrt{\bar{\alpha}_{t_\delta}}$ within the denoiser we have

$$
\mathbf{z}_{0,h} \approx D_{(1-\bar{\alpha}_{t_\delta})/2}(\sqrt{\bar{\alpha}_{t_\delta}}\overline{\mathbf{x}_{0,s}} + \sqrt{1 - \bar{\alpha}_{t_\delta}}\epsilon).
$$

Applying this kernel iteratively, approximating the MMSE denoiser $D_{(1-\bar{\alpha}_{t_\delta})/2}$ by the score-based prior $\bar{G}_\theta(\cdot, t_\delta)$, we obtain the updates

$$
\begin{aligned}
\bar{\mathbf{x}}_0^{(k+1)} &= \mathbb{E}[\mathbf{x}|\mathbf{y}, \mathbf{z} = \mathbf{z}^{(k)}, \delta] \\
&= \mathrm{Prox}_{\delta g_y}(\mathbf{z}^{(k)}) \\
\mathbf{z}^{(k+1)} &= \bar{G}_\theta(\sqrt{\bar{\alpha}_{t_\delta}}\bar{\mathbf{x}}_0^{(k+1)} + \sqrt{1 - \bar{\alpha}_{t_\delta}}, t_\delta),
\end{aligned}
$$

where we relate the conditional mean of $\mathbf{x}$ given $\mathbf{y}$ and $\mathbf{z}$ to the proximal operator of $g_y$ since $p(\mathbf{x}|\mathbf{z})$ defines a conjugate Gaussian prior for $p(\mathbf{x}|\mathbf{y}, \mathbf{z})$. This recovers the LATINO algorithm equation 5 under the mapping $\sqrt{\bar{\alpha}_{t_\delta}}\bar{\mathbf{x}}_0^{(k+1)} + \sqrt{1 - \bar{\alpha}_{t_\delta}} = \mathbf{x}_{t_\delta}^{(k+1)}$ and $\mathbf{z}^{(k+1)} = \mathbf{x}_0^{(k+1)}$. The kernel equation 5 can thus be understood as a proximal split-Gibbs timestep for the Langevin diffusion with a score-based plug-and-play prior. A closely related approach was employed in Mbakam et al. (2024) based an explicit gradient discretisation of the marginal diffusion Equation 17 and using a small number of DDIM iterations to formulate a stochastic plug-and-play denoising prior. Following Pereyra et al. (2023); Laumont et al. (2022); Martin et al. (2025 (to appear), the parameter $\delta$ (equivalently, $t_\delta$) can be tuned to provide asymptotic convergence to the posterior distribution. However, as observed empirically in Appendix D.1, when applied as a zero-shot method the MCMC chain arising from the LATINO updates requires a significant number of NFEs to reach stationarity. Following the successful unfolding of optimization procedures for imaging illustrated in Monga et al. (2021), we aim in this work to generate accurate posterior samples unfolding a small number $K$ LATINO iterations as a likelihood-aware neural network. By designing a network in this manner, we obtain a modular and interpretable model with a natural embedding of the forward observation model.

In Spagnoletti et al. (2025), the same algorithm is derived for large time approximations $h \approx 1$. This derivation relies on the use of a pre-distilled consistency model prior to construct a kernel which contracts to the same invariant measure as equation 17. A key difference for the proposed approach in Section 3 is that we start with a diffusion model designed for small time-steps. Several iterations of LATINO are then distilled to construct a conditional consistency model for a determined class of image restoration problems.

## C    Implementation details

### C.1    Models

In this work, we used two distinct pre-trained models for our experiments. The first model is derived from ADM (Dhariwal & Nichol, 2021), originally pre-trained on the ImageNet dataset at the resolution of $256\times256$. The second model is based on a pre-trained model from DDPM (Ho et al., 2020), which was pre-trained on LSUN dataset at the resolution of $256\times256$. The architectures of these models differ significantly and a detailed comparison of their architecture, parameter counts and computational requirements is provided in Table 8.

Regarding LoRA configuration, we implemented exclusively on the attention layers of the pre-trained model to improved its adaptability for specific image restoration tasks while reducing the computational overhead.

Table 8: Models summarize: $|\theta|$ denotes the total number of frozen parameters, $|\Delta_\theta|$ indicates the total LoRA parameters and $|\psi|$ represents the total RAM parameters. **Model 1** refers to the ADM pre-trained on the ImageNet dataset, while **Model 2** denotes the DDPM pre-trained on the LSUN dataset.

| Models | rank($\Delta_\theta$) | #Attn layers | $|\theta|$ | $|\Delta_\theta|$ |
|---|---|---|---|---|
| **Model 1** | - | 16 | 552814086 | 0 |
| **Model 2** | - | 5 | 113673219 | 0 |
| **Model 1 w/ LoRA** | 5 | 16 | 552814086 | 414720 |
| **Model 2 w/ LoRA** | 5 | 5 | 113673219 | 122880 |
| **RAM** | - | 0 | 35618953 | 0 |

### C.2    Training procedure

We adopted a training procedure that is carefully designed to ensure robust performance across tasks such as deblurring, inpainting and super-resolution. The training objective integrates reconstruction and adversarial losses to enhance sample generation quality. For the reconstruction loss, we used a consistency term to enforce data fidelity and a LPIPS term to enhance perceptual quality. Conversely, the adversarial term is designed to promote the model to generate realistic sample. Additionally, the training loss incorporates gradient penalty term to stabilize training by enforcing Lipschitz continuity on the discriminator. To facilitate reproducibility, comprehensive details regarding batch size ($bs$), learning rate ($lr$), optimizer and specific weights for loss term are summarized in Table 9.

Table 9: **Model 1** refers to the ADM pre-trained on the ImageNet dataset, while **Model 2** denotes the DDPM pre-trained on the LSUN dataset.

| | $lr$ | $bs$ | optimizer | weight decay | $\omega_{\ell_2}$ | $\omega_{GP}$ | $\omega_{GS}$ |
|---|---|---|---|---|---|---|---|
| **Model 1** | $1e^{-4}$ | 2 | AdamW | 0.01 | 1 | 0.1 | 0.01 |
| **Model 2** | $1e^{-4}$ | 4 | AdamW | 0.01 | 1 | 0.1 | 0.01 |
| **RAM**$_\psi$ | $1e^{-5}$ | - | AdamW | 0.01 | - | - | - |

### C.3    Setting experiments

**Box inpainting**    We consider a fixed and centered square mask of size $128 \times 128$ pixels, $K = 5$ and $\sigma = 0.025$.

**Random inpainting**    In this experiment, a random binary mask is applied with missing pixel rates of 80% and 70%. We use $K = 3$ iterations for algorithm unrolling and a noise level of $\sigma = 0.025$.

**Super-resolution:** For SR experiments with a scaling factor $m$, images from the LSUN dataset are downsampled by convolving with a bicubic filter of size $m^2 \times m^2$. In contrast, for the ImageNet dataset, images are first convolved with a $3 \times 3$ Gaussian kernel[4] and bandwidth of 10, followed by a downsampling operation with factor $m$. Unless otherwise stated, we consider downsampling factor $m = 4$. The number of iterations in algorithm unrolling is set to $K = 3$, and experiments are conducted with noise levels $\sigma \in \{0.01, 0.025\}$.

**Deblurring** For this experiment, consider 3 types of blur.

- **Gaussian blur**: On Imagenet experiments, we consider kernel sizes of $5 \times 5$ and $25 \times 25$ pixels and a bandwidth of 10. For the LSUN bedroom dataset, we use consider the anisotropic Gaussian blur used in Garber & Tirer (2025); Kawar et al. (2022) of size $9 \times 9$ and with bandwidth $(20, 1)$.

- **Motion blur**: We generate random motion blur kernels with kernel sizes of $5 \times 5$ using the approach described in Schuler et al. (2016)[5],

- **Uniform blur** with kernel sizes of $9 \times 9$.

We conduct experiments under noise levels $\sigma \in \{0.01, 0.025, 0.05\}$ and using $K = 3$ iterations for algorithm unrolling.

**JPEG restoration** For this experiment, we applied JPEG compression with a quality factor of **QF=10** and introduced Gaussian noise with variance $0.01^2$.

### C.4 Memory and time consumption

Table 10 presents a comparative analysis of memory consumption and computational time for the models evaluated in this work. The results demonstrate that fine-tuning pre-trained models using LoRA technique reduces GPU memory usage and inference time. The table also includes the memory usage and inference time for RAM.

For the JPEG restoration task, we obtained an inference time of **4.01** seconds. This increased computational time is because the proposed method requires a sub-iterative method to solve the proximal step in Figure 2.

Table 10: Models summarize: $K$ denotes the number of unfolding steps. **Model 1** refers to the ADM pre-trained on the ImageNet dataset, while **Model 2** denotes the DDPM pre-trained on the LSUN dataset.

| Models | K | GPU (GB) | Time (s) | Resolution |
|---|---|---|---|---|
| **Model 1** | - | 16.49 | - | $256^2$ |
| **Model 2** | - | 3.39 | - | $256^2$ |
| **Model 1 w/ LoRA** | 3 | 0.013 | 0.89 | $256^2$ |
| **Model 2 w/ LoRA** | 3 | 0.004 | 0.53 | $256^2$ |
| **RAM** | 1 | 1.06 | 0.31 | $256^2$ |

## D  Additional results

In this appendix, we provide additional qualitative results between $UD^2M$ applied to the problems in Section 4. Qualitative comparisons to similar methods for deblurring and super-resolution on ImageNet may be found in Figures 12 and 13. Qualitative reconstructions obtained via the universal $UD^2M$ scheme discussed in Section 4.3.6 can be found in Figure 14.

---

[4]The Gaussian kernel act as a low-pass filter to reduce aliasing before downsampling.

[5]An implementation of this operator can be found in Deepinverse.

Additional qualitative comparisons of UD$^2$M (with RAM) on the LSUN bedroom validation set can be found in Figures 15 and 16 for the gaussian deblurring and random inpainting tasks, each with in-distribution noise level $\sigma = 0.025$. To illustrate qualitatively the extensibility of the unfolded architecture to out-of-distribution noise, additional comparisons are presented in Figure 17 using noise level $\sigma = 0.05$ which was not used in training UD$^2$M. For the comparison methods on LSUN bedroom, we use the reconstructed images provided in Garber & Tirer (2025). To illustrate the generative capabilities of the UD$^2$M prior, additional reconstructions for box inpainting on LSUN bedroom are provided in Figure 18

### D.1 Ablation Study on MNIST

We consider further ablation studies for inverse problems on MNIST digits, with 60,000 training and 10,000 test images. For ease of computation, we add a $2 \times 2$ zero-padding to resize each image to shape $32 \times 32$. The small size of these images allows for efficient training of several models for extensive ablation studies and for many posterior samples to be generated, allowing for accurate exploration of the learned posterior data. We test UD$^2$M on a challenging inverse problem on this data, to benchmark the convergence of our method for a small number of NFEs.

**Model**  For our deep unfolded UD$^2$M architecture, we use a score-based denoiser with the same architecture as Dhariwal & Nichol (2021). The score-based denoiser is downsized to take as input grayscale images of size $32 \times 32$. We pre-train the score model to generate images from the $32 \times 32$ MNIST dataset using 220,000 training iterations with a batch-size of 64. The final model contains 16,761,409 parameters.

Through adversarial distillation, we fine-tune the score-model into a conditional consistency model, generating conditional samples of $x_0$ given $x_t$ and an observation $y$ by unfolding $K$ iterations of the LATINO kernel. To compare models with different number of unfolded iterations, we train 4 separate models with $K = 1, 2, 4$ and $K = 8$, respectively. For each model, we use the same approach as above and freeze the weights in the pre-trained score and apply a learnable low-rank correction only to weights within the attention layers. We apply LoRA with rank $r = 5$, resulting in 65,280 trainable weights representing 0.39% of the original model.

**Experiment**  We consider the SR ($\times 4$) problem, using additive Gaussian noise with variance of size $\sigma^2 = 0.05^2$. Due to the small size of MNIST digits, down-sampling by a factor of 4 represents a significant degradation in the quality of the image (see 10). By considering a highly ill-conditioned problem, we aim to highlight the efficiency of UD$^2$M to converge to an accurate posterior distribution with only a small number of NFEs. This allows an effective demonstration of the enhanced convergence rate of the unfolded LATINO iterates over $K = 8$ iterations, which is in contrast to the large $K \gg 1$ number of iterations typically required for MCMC sampling (Pereyra, 2016).

**Comparison**  To benchmark against a state-of-the art zero-shot MCMC sampler, we compare our method to the long-term dynamics of the LATINO kernel equation 5. Using the same pre-trained denoiser as discussed above, we run $K \approx 50,000$ iterations of the LATINO algorithm. To evaluate the performance gain by embedding LATINO within a distilled diffusion loop through UD$^2$M, we do not apply fine-tuning or distillation to the kernel. Instead, interpreting LATINO as a plug-and-play discretisation of the Langevin diffusion, we tune the step-size by hand using the theoretical bounds in Laumont et al. (2022).

**Evaluation Metrics**  For the evaluation, we consider the PSNR and LPIPS metrics used above. The FID metric implicitly assumes that the Inception-v3 network accurately maps the prior distribution to a Gaussian in the encoding space. Due to limitations with the Inception-v3 network mapping MNIST images to a Gaussian distribution, we replace this network with a VAE encoder $\mathcal{E}_{\text{MNIST}}$ pre-trained explicitly in Holden et al. (2022) to encode the MNIST dataset into a 12-dimensional Gaussian distribution. The Frechet distance is then computed between VAE-encoded MNIST digits and VAE-encoded digits reconstructed through our model. Following Bendel et al. (2023), we compute posterior approximation quality using the conditional Frechet inception distance (CFID) Soloveitchik et al. (2021). For approximations $\hat{x}$ of the ground truth $x$, the CFID computes the conditional Frechet distance between the embeddings $p(\mathcal{E}_{\text{MNIST}}(x)|\mathcal{E}_{\text{MNIST}}(y))$ and $p(\mathcal{E}_{\text{MNIST}}(\hat{x})|E_{\text{MNIST}}(y))$, averaged over many observations $y$.

| K\N | 1 | 2 | 4 | 8 | 16 |
|---|---|---|---|---|---|
| 1 | 14.8 | 15.5 | 15.5 | 15.7 | 15.9 |
| 2 | 16.3 | 17.1 | 17.3 | 17.5 | 17.5 |
| 4 | 18.1 | 18.2 | 18.2 | 18.2 | 18.2 |
| 8 | 18.5 | 18.6 | 18.6 | 18.6 | 18.6 |

(a) PSNR

| K\N | 1 | 2 | 4 | 8 | 16 |
|---|---|---|---|---|---|
| 1 | 0.18 | 0.28 | 0.22 | 0.14 | 0.11 |
| 2 | 0.07 | 0.10 | 0.07 | 0.05 | 0.05 |
| 4 | 0.05 | 0.04 | 0.04 | 0.04 | 0.04 |
| 8 | 0.04 | 0.04 | 0.04 | 0.04 | 0.04 |

(b) LPIPS

| K\N | 1 | 2 | 4 | 8 | 16 |
|---|---|---|---|---|---|
| 1 | 4.87 | 3.82 | 3.44 | 2.86 | 2.49 |
| 2 | 2.81 | 1.86 | 1.16 | 0.78 | 0.73 |
| 4 | 0.31 | 0.24 | 0.24 | 0.23 | 0.22 |
| 8 | 0.15 | 0.15 | 0.15 | 0.15 | 0.15 |

(c) FID

| K\N | 1 | 2 | 4 | 8 | 16 |
|---|---|---|---|---|---|
| 1 | 6.91 | 5.16 | 4.83 | 4.08 | 3.53 |
| 2 | 3.65 | 2.25 | 1.47 | 1.01 | 0.94 |
| 4 | 0.47 | 0.38 | 0.37 | 0.36 | 0.35 |
| 8 | 0.22 | 0.22 | 0.22 | 0.22 | 0.22 |

(d) CFID

Table 11: Comparison of reconstruction metrics for UD$^2$M models trained on the MNIST dataset for a range of unfolded steps $K$ and DDIM steps $N$. For each observation $y$, the PSNR is computed between the average of 16 samples from the estimated posterior distribution. Since the remaining metrics quantify sample quality, they are computed between a single posterior sample and the ground truth.

**Results** We train 4 instances of the unfolded LATINO architecture, with $K = 1, 2, 4, 8$ respectively. For each model, we run the unfolded UD$^2$M sampler for $N = 1, 2, 4, 8, 16$ iterations for a total of $NK$ NFEs. The evaluation metrics for each instance are shown in Table 11. For the PSNR, we compare the ground truth to an average of 16 UD$^2$M samples to approximate the MMSE estimator for the problem. The remaining metrics use a single posterior sample. For each metric, we notice fast convergence within $K \approx 4$ iterations. For this ill-conditioned problem, we find a small improvement by additionally using $K = 8$ unfolded steps. Qualitative results are shown for a sample observation in Figure 10. For each $K$, $N$ pair, we plot a single UD$^2$M sample, along with the mean and standard deviation estimated from 16 UD$^2$M samples. The standard deviation plots are amplified by a factor of $3\times$ to facilitate visual comparison and align with approximately 99% credible intervals. For each shown standard deviation, we indicate in the upper left corner the correlation to the corresponding absolute error. We observe sharper images and greater sample diversity for $K \geq 4$. Using $K = 8$, we observe around 10% increase in correlation between the posterior variance and the absolute error. In each instance, deviations of individual samples from the sample mean illustrate the ability of our method to avoid mode collapse in the learned posterior distribution, providing a wide exploration of viable samples. Reference reconstructions are shown for the long-time average of the zero-shot LATINO algorithm over 50,000 NFEs. The convergence of LATINO to an invariant measure is summarized in Figure 11 (left) For our unrolled model, we observe similar qualitative performance to LATINO with $K \geq 4$ iterations with $16KN \ll 50,000$ NFEs to compute the mean and standard deviation, along with better correlation of the standard deviation to the true error. In all metrics and qualitative comparison, we observe a greater benefit from increasing $K$ opposed to $N$. This highlights the effectiveness of unfolding a small number of LATINO iterations to distill the conditional diffusion model to require only a small number of iterations.

To benchmark the uncertainty quantification properties of UD$^2$M from accurate posterior coverage, we plot empirical coverage probabilities in Figure 11 (right). To focus on important search directions, we compute the empirical coverage on the latent space of the encoder $\mathcal{E}_{\mathrm{MNIST}}$. The coverage is computed for $K = 1, 2, 4, 8$ unfolded iterations, with $N$ scaled to a fixed number of 16 NFEs. For this calculation, following Tachella & Pereyra (2023); Cherif et al. (2024), we improve sample diversity in directions corresponding to small eigenvalues of the super-resolution operator by bootstrapping our reconstructed samples through equivariant transformations of the reconstructions as in Cherif et al. (2024). We notice that increasing $K$ has a positive impact on the empirical coverage probabilities, with accurate results for $K \geq 4$. This highlights the usefulness of UD$^2$M as a tool for accurate uncertainty quantification.

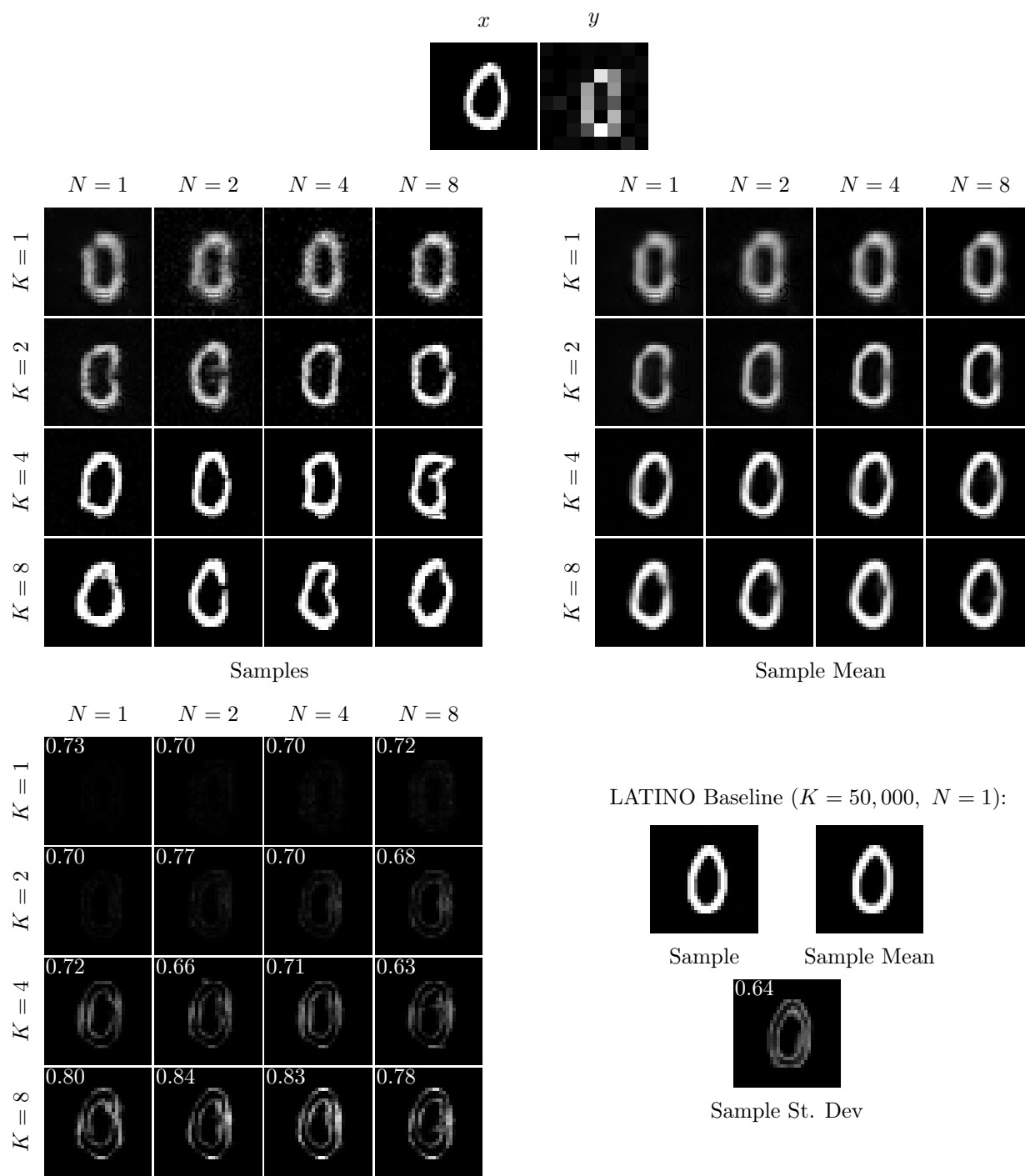

Figure 10: Example reconstructions for the MNIST SR (×4) problem for $K, N \in \{1, 2, 4, 8\}$. **Top:** The ground truth image $x$ and observation $y$. **Middle**: Samples (left) and mean estimate (right) from the learned UD²M posterior distribution with $K$ unfolded LATINO iterations using $N$ DDIM steps. **Bottom**: The corresponding posterior standard deviation, computed using 16 conditionally independent samples given y. The correlation of the standard deviation to the absolute error is indicated in the upper left of each standard deviation plot. A reference mean and standard deviation of the zero-shot LATINO kernel after 50,000 iterations are shown in the bottom right.

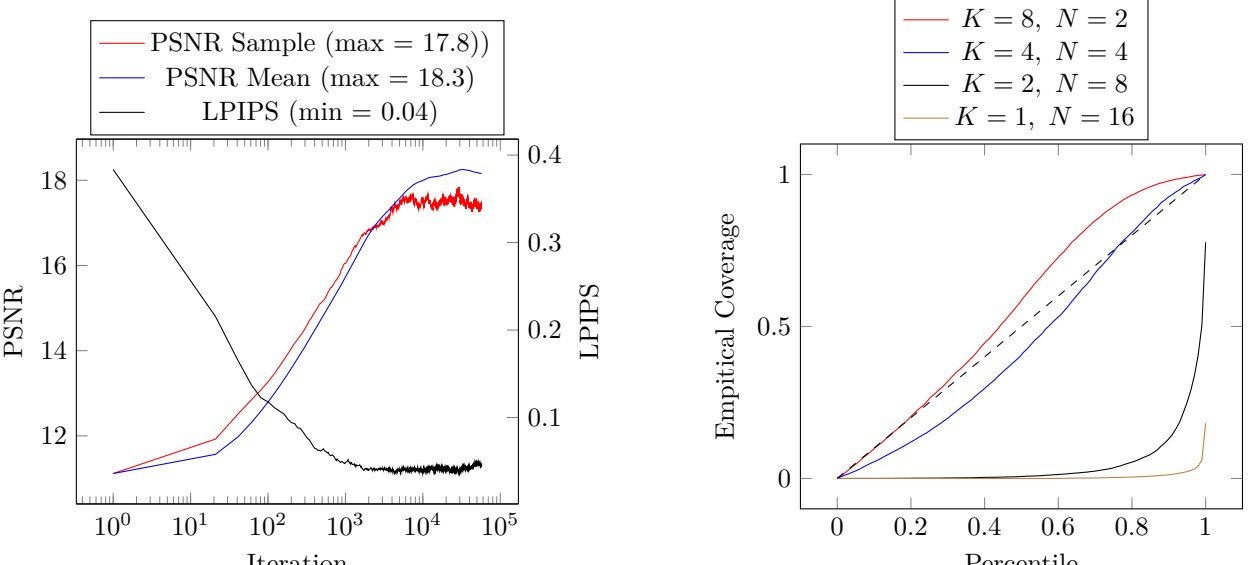

Figure 11: **Left:** Empirical convergence of the zero-shot LATINO algorithm against number of iterations. The PSNR of a single sample and of the Markov Chain mean are shown. Additionally, the LPIPS of a single posterior sample is shown. Results are averaged over 32 independent test images. **Right:** Empirical coverage probabilities for $\text{UD}^2\text{M}$ for $K = 1, 2, 4, 8$ and $N$ scaled to fix a total of 16 NFEs. The dashed line represents perfect posterior coverage.

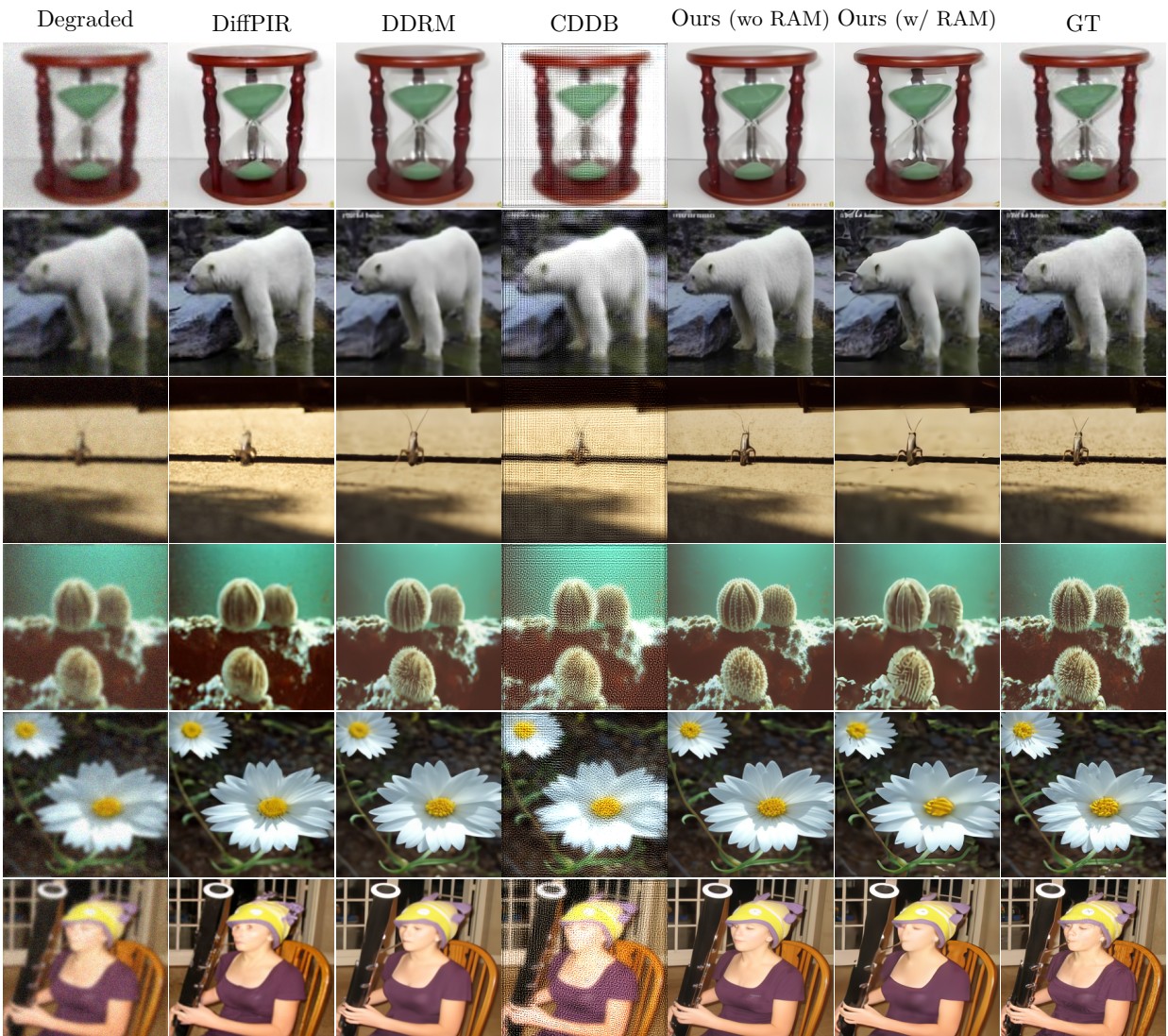

Figure 12: Examples of posterior samples for the task Gaussian deblurring on ImageNet with noise level $\sigma = 0.05$ and Gaussian kernel bandwidth of 10.

Degraded  DiffPIR  DDRM  CDDB  Ours (wo RAM) Ours (w/ RAM)  GT

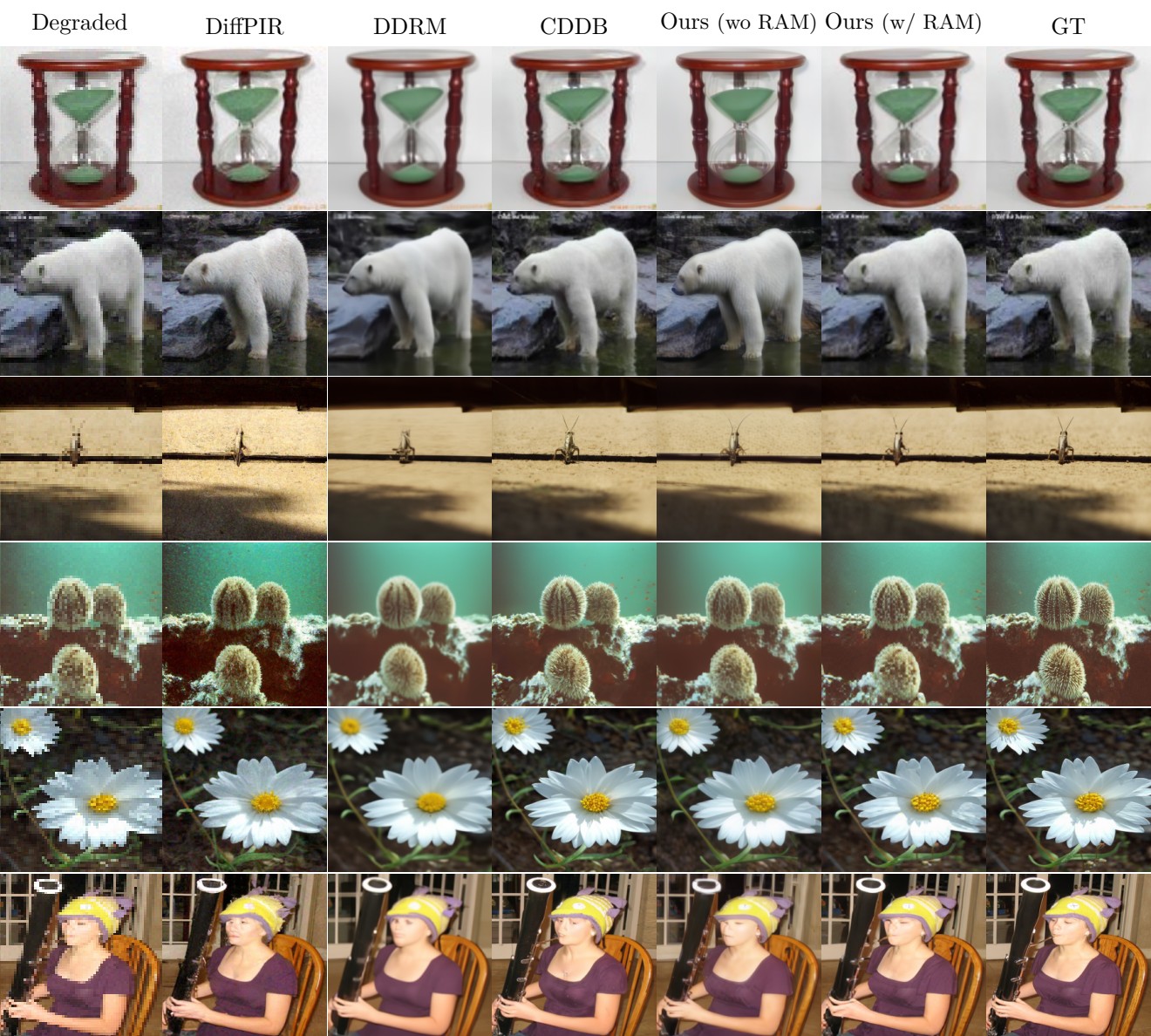

Figure 13: Examples of posterior samples for the task SR($\times$4) with noise level $\sigma = 0.05$ on ImageNet 256
.

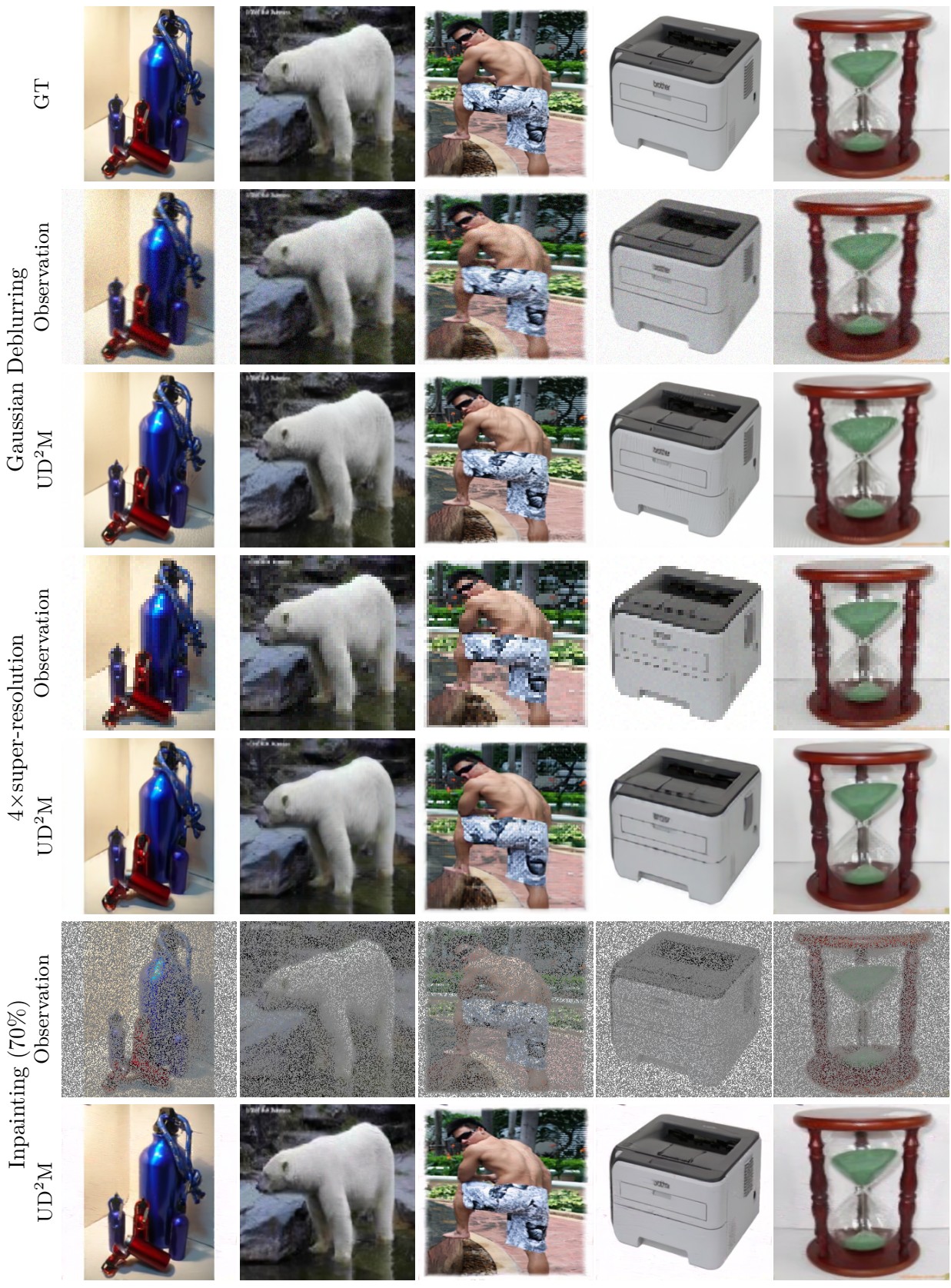

Figure 14: Examples of posterior samples generated by a "universal" UD$^2$M sampler on the following tasks on ImageNet 256: Gaussian Deblurring with a kernel bandwidth of 10, random inpainting (70%), super-resolution (×4). Noise level $\sigma = 0.01$.

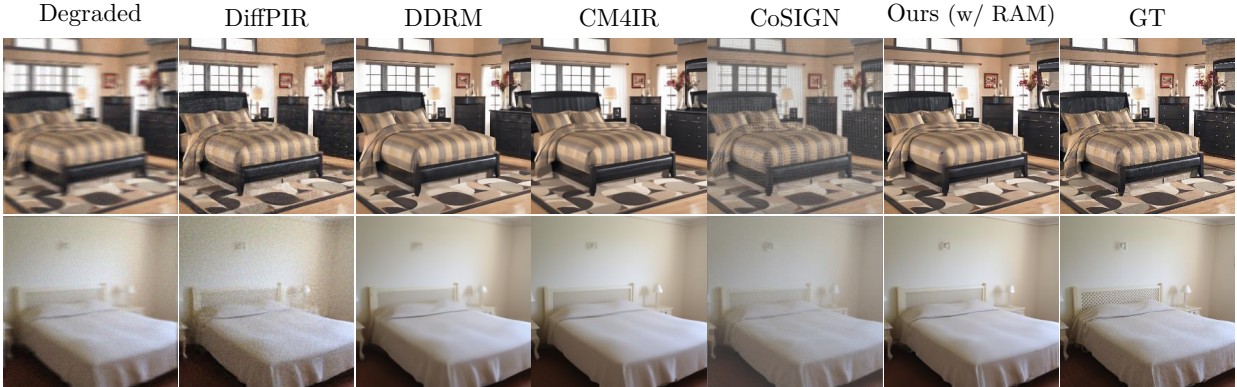

Figure 15: Qualitative results from the LSUN bedroom validation set applied to anisotropic Gaussian deblurring with bandwidth $(20, 1)$ and additive Gaussian noise with standard deviation $\sigma = 0.025$.

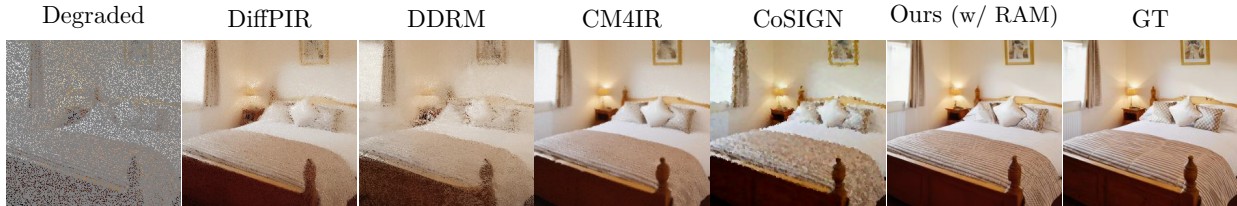

Figure 16: Qualitative results from the LSUN bedroom validation set applied to random inpainting with 80% masked pixels and additive Gaussian noise with standard deviation $\sigma = 0.025$.

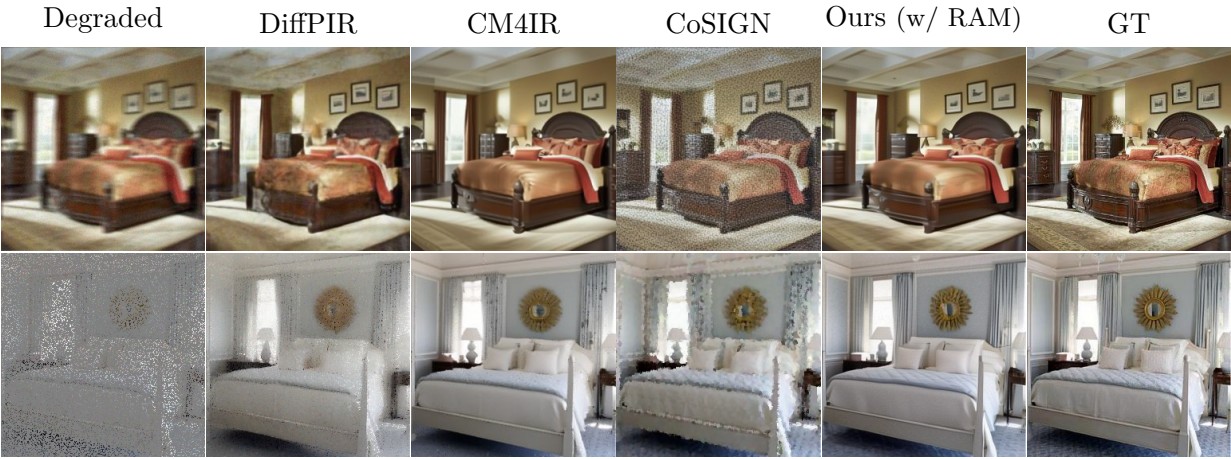

Figure 17: Qualitative results from the LSUN bedroom validation set applied to anisotropic Gaussian deblurring with bandwidth $(20, 1)$ (top) and random inpainting with 80% masked pixels (bottom). Each with additive Gaussian noise with standard deviation $\sigma = 0.05$, which is out of the training distribution for our model which was trained for $\sigma = 0.025$.

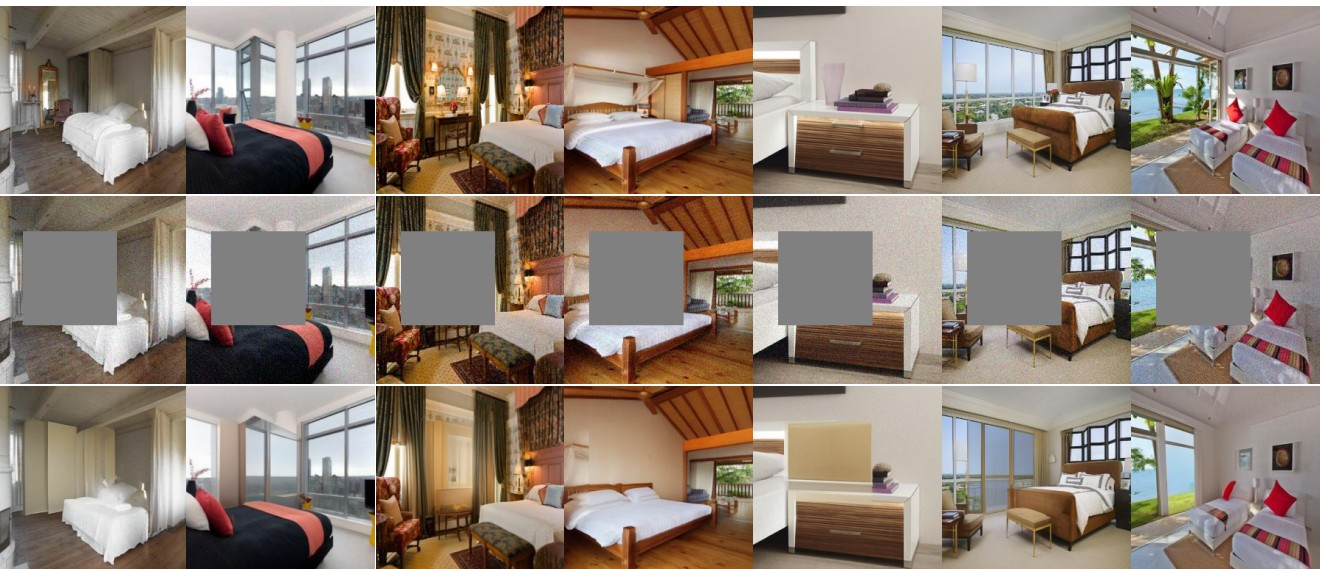

Figure 18: Examples of posterior samples for the task box inpainting with noise level $\sigma = 0.05$ on LSUN Bedroom. From top to bottom: Ground truth, degraded observation, UD$^2$M reconstruction (without RAM).

