# OpenReview forum: "Learning few-step posterior samplers by unfolding and distillation of diffusion models"
_TMLR — Accepted by TMLR_

### Review · Reviewer_qiWz · 2025-08-20

**Summary Of Contributions:**

Summary Of Contributions:

The paper has as an overarching goal the task of learning few-step samplers that draw from the posterior for imaging problems that are severely ill-conditioned or ill-posed.
The paper proposed the Unfolded and Distilled Diffusion Model method, which unfolds K iterations of the LATINO Langevin posterior sampler inside a conditional diffusion pipeline and then distils to a few-step consistency model. The proposed approach uses likelihoods via proximal operators and supports joint training over a family of likelihoods. Experiments on ImageNet‑256 and LSUN Bedroom across Gaussian deblurring, inpainting, superresolution, and restoration of JPEG compression artifacts report strong FID/LPIPS.


Strengths And Weaknesses:

Strengths:

- Clear and well-written
- The Unfolded and Distilled Diffusion Model method is well presented. As far as my knowledge, the authors claim of this being the first deep unfolding of an MCMC sampler within diffusion‑model imaging is correct.
- Exact likelihood use via a proximal step inside Langevin avoids common likelihood approximations in DM‑based posterior sampling.
- Demonstrate decent empirical evidence across ImageNet/LSUN tasks for multiple metrics, ablations useful for key hyperparameters (choice of rank for LoRA, number of sampling steps, etc). Sufficient details given for reproducibility.


Weaknesses:

- My main issue is no results explicitly assess posterior validity. The paper does a good job across various metrics typically used to assess image quality and diversity for image tasks; however, if want to present this as posterior sampling, I would like to see something more concrete. For instance, to verify details in the training objective don't lead to an overconfident posterior.
- No comparison to LATINO
- Proximal step solving. Reliance on sub‑iterative solvers for some tasks (e.g., non‑linear cases); Appendix A is helpful and many problems may admit fast proximal solutions, but when sub‑iterations are required inference slows.

**Audience:**

Yes

**Audience Explanation:**

Yes. In particular, this paper should appeal to readers interested in Bayesian computational imaging.

**Claims And Evidence:**

Yes

**Claims Explanation:**

Yes. Overall, the main contributions are novel and the method's performance is reasonably supported by decent experimentation.

**Requested Changes:**

Requested Changes:
- My main requested change is regarding posterior validity. Either include some additional posterior validity checks (some options might be: simulation-based calibration, assess empirical coverage of credible sets, posterior predictive log score, run on toy example and compare with golden standard method) or some softening of claims surrounding posterior sampling.

Other optional changes:
- LATINO, a direct baseline would clarify how much the proposed approach improves on LATINO.

Spelling and grammar:
- “we are particularly interest in imaging problems” ->  interested
- “for solving for inverse problems” -> for solving inverse problems
- “out method is not as efficient” -> our
- “LSUN bederoom dataset” -> bedroom
- “Ground Thruth” -> truth

---

> ### Author Response · Authors · 2025-10-18
> **Response to Reviewer qiWz**
>
> Thank you for carefully reading our manuscript and for your constructive feedback. In response to your requested changes, we have added an extra ablation summary, summarised below, to validate the accuracy of the learned posterior samples from our model. Thank you for pointing out several typos and grammatical errors. These have all been corrected.
>
> **My main requested change is regarding posterior validity. Either include some additional posterior validity checks (some options might be: simulation-based calibration, assess empirical coverage of credible sets, posterior predictive log score, run on toy example and compare with golden standard method) or some softening of claims surrounding posterior sampling.**
>
> We thank the reviewer for highlighting this important point, which was insufficiently addressed in the original manuscript. To address this comment, we have added a new experiment in Appendix D.1 on MNIST super-resolution, where performance metrics can be accurately estimated through extensive repetitions. The key motivation for using this dataset is that MNIST digits can be reliably encoded as Gaussian parameters in a d=12 latent space, enabling a meaningful comparison between the Fréchet distances of ground-truth and reconstructed images. Owing to the small image size, we can efficiently draw numerous posterior samples to estimate coverage probabilities precisely. These results, shown in Figure 11, demonstrate accurate coverage for K=4 unfolded steps.
>
> **LATINO, a direct baseline would clarify how much the proposed approach improves on LATINO.**
>
> As part of the experiment in Appendix D.1, we compare to a long-time average of the zero-shot LATINO scheme. We observe comparable performance for UD2M with significantly reduced NFEs. This highlights the effectiveness of our method at reducing the sampling cost considerably.

---

### Review · Reviewer_xFEp · 2025-08-23

**Summary Of Contributions:**

The paper presents UD$^2$M, a framework for enabling pre-trained Diffusion Models (DMs) to perform posterior sampling, in particular in the case of inverse problems in computational imaging. The paper builds upon LATINA, which distills a DM to a Consistency Model (CM), and uses it as a prior in combination with a proximal algorithm with respect to the likelihood of the conditioning variable $y$. The authors add many techniques, such as Algorithm Unfolding of the LATINA Langevin sampling steps, LoRa to make these steps fine-tunable, and adversarial training with a discriminator.

Strengths:
1. The paper is generally well written, and most parts are clear
2. There are interesting relations with other parallel works in the literature
3. There is an extensive evaluation and ablation studies

Weaknesses:
1. There are quite a lot of modules and techniques, which I feel don't get presented in a very clear order. For example, what is $L_\vartheta$ concretely? What are its inputs? Why are parameters $\phi$ introduced before the discriminator $D_\phi$ in section 3.2?
2. Although text passages are fairly easy to follow, the math notation is quite hard to understand at times. I would advise a notation (and definitions) section.
3. I feel like the line between the related works and the proposed method is blurry in the writing. For someone not familiar with each specific subject, it is difficult to follow what is proposed and what already exists. I advise a paragraph (preferably with a list) explaining the contributions of the paper, and what distinguishes it from the related works.

Questions:
1. I found the sentence "Importantly, unlike zero-shot methods, here we consider that the likelihood function $p(y|x_0)$ associated to $p(x_0|y, x_t)$ belongs a class of likelihood functions of interest that is fixed during the training of $L_\vartheta$, while retaining the flexibility to specify the exact parameters of the likelihood during inference time" to be a bit confusing. Isn't it fixed also for other methods?
2. To clarify, is the idea of unfolding LATINO to have a fixed number of specialized layers, fine-tuned with LoRa, instead of having equal "recurrent" steps?

**Additional Comments:**

Typos:
- Spangnoletti -> Spagnoletti
- Introduction:
	- Within this context, we are particularly "interest" -> interested

There might be other typos after that point, so I advise a full paper check.

**Audience:**

Yes

**Audience Explanation:**

The paper advances the field of Diffusion Models, which is interesting and a hot topic in the current state of the Machine Learning field.

**Broader Impact Concerns:**

The Broader Impact Statement provided in the paper addresses fully the possible concerns.

**Claims And Evidence:**

Yes

**Claims Explanation:**

The authors demonstrated empirically that UD$^2$M has high performance with flexibility and a low number of NFE, making it efficient.

**Requested Changes:**

Critical changes are those explained in Weaknesses 1-3. In general, I advise making the paper more accessible to readers not familiar with every single technique or their usefulness.

---

> ### Author Response · Authors · 2025-10-18
> **Response to Reviewer xFEp**
>
> Thank you for your careful reading of the paper and for providing constructive feedback regarding our manuscript. We have made several modifications to the text in order to clarify the contributions, notation and setup in response to the raised questions. We address each of your points below:
>
> **There are quite a lot of modules and techniques, which I feel don't get presented in a very clear order. For example, what is $L_\vartheta$ concretely? What are its inputs? Why are parameters $\phi$ introduced before the discriminator $D_\phi$ in section 3.2?**
>
> We have made several changes, highlighed in red text, in the revision with the intention of improving the clarity of the document. In response to the specific points raised, we have added more description of $L_\vartheta$ and $\phi$ before they are used in the text.
>
> **Although text passages are fairly easy to follow, the math notation is quite hard to understand at times. I would advise a notation (and definitions) section.**
>
> **I feel like the line between the related works and the proposed method is blurry in the writing. For someone not familiar with each specific subject, it is difficult to follow what is proposed and what already exists. I advise a paragraph (preferably with a list) explaining the contributions of the paper, and what distinguishes it from the related works.**
>
> We thank you for these suggestions. We have added both of the suggested sections at the end of the introduction.
>
> **I found the sentence "Importantly, unlike zero-shot methods, here we consider that the likelihood function $p(y|x_0)$ associated to $p(x_0|y, x_t)$ belongs a class of likelihood functions of interest that is fixed during the training of $L_\vartheta$, while retaining the flexibility to specify the exact parameters of the likelihood during inference time" to be a bit confusing. Isn't it fixed also for other methods?**
>
> The intended implication is that by training our model to a specific class of likelihood functions, we specialise our model weights to restore images observed through forward observation models which are derived from the pre-determined class of likelihood functions at training time. This thereby restricts the class of likelihood functions our model is trained for at inference time to those that are similar to the training data. In comparison, zero-shot algorithms are designed to work for any likelihood function at inference time. As seen in the numerical results, specialising to a determined class of likelihood models provides significant improvements to restoration quality and efficiency at inference time.
>
>
> **To clarify, is the idea of unfolding LATINO to have a fixed number of specialized layers, fine-tuned with LoRa, instead of having equal "recurrent" steps?**
>
> To formalise precisely that LATINO is a recurrent approximation of the Langevin diffusion, we have added a detailed derivation of LATINO in Appendix B. The purpose of our work is to take a small number of LATINO iterations to construct a recurrent neural-network (inspired by the Langevin diffusion), and learn the LoRA and discretisation weights to obtain a distilled conditional score for the determined reconstruction metric. The benefits of this are to obtain an algorithm which converges much faster than standard LATINO without a distilled score. We have added a comparison with a vanilla LATINO algorithm without a pre-distilled consistency model to Figure 11 (left) to illustrate this point. In addition, by conditioning on $x_t$ within each UD2M module, we obtain a model which is well-conditioned and interpretable, that compares favourably to similar distilled diffusion models for image restoration.
>
>
> Thank you for carefully pointing out various typos, we have fixed them all.

---

### Review · Reviewer_G1t8 · 2025-09-06

**Summary Of Contributions:**

The paper proposes UD2M, a few-step posterior sampler for image restoration that
1. unrolls a small number of LATINO Langevin iterations into a trainable block,
2. uses an explicit proximal data-consistency step to encode the likelihood, and
3. plugs in a distilled diffusion prior with lightweight LoRA adapters.

Chaining a handful of such blocks across decreasing diffusion noise levels yields fast, good-quality reconstructions. Experiments on deblurring, inpainting, ×4 super-resolution, and JPEG (ImageNet/LSUN) show competitive or improved FID/LPIPS/PSNR at roughly order-10 NFEs, with ablations on LoRA rank and step budgets.

Strengths:
1. Clear separation of physics (prox) and prior (distilled DM); compelling few-step efficiency.
2. Broad empirical study across tasks/datasets; strong results for the reported NFE budgets.
3. Practical design (LoRA, closed-form prox for linear-Gaussian cases) that keeps training and inference lightweight.

Weakness:
1. Requires task-specific adaptation; prox for non-linear operators relies on approximate solves.
2. The paper empirically studies the effect of unrolling, showed good results, but provides no non-asymptotic guarantee for the learned $K$-step unrolled map vs. full LATINO iterations.
3. The trade-off between inner $K$ and outer $N$ under a fixed budget is not isolated.

**Audience:**

Yes

**Audience Explanation:**

TMLR readers working on diffusion models, computational imaging, and deep unrolling would find this relevant because it combines an explicit likelihood prox with a LoRA-adapted few-step diffusion prior for fast posterior sampling.

**Claims And Evidence:**

Yes

**Claims Explanation:**

Empirical evidence is strong. Evidences provided across deblurring, inpainting, ×4 super-resolution, and JPEG on ImageNet/LSUN the method reports competitive/improved FID/LPIPS/PSNR with ablations (LoRA rank, step budgets) and noise-level generalization.

However, the theoretical support is limited, with no non-asymptotic bounds for the unrolled sampler vs. full LATINO, and only heuristic treatment of inexact proximal steps.

**Requested Changes:**

1. Provide a error bound between the learned $K$-step unrolled map vs. full LATINO iterations under mild assumptions. If relevant theorems exist, include or discuss them. Or at least analyze and discuss the finite-step error and its effect on the task.
2. Provide a $(K,N)$ grid under fixed NFE showing the best allocation.

---

> ### Author Response · Authors · 2025-10-18
> **Response to Reviewer G1t8**
>
> Thank you for your careful reading of the paper and detailed feedback regarding our manuscript.
>
> In response to your requested changes, we have made the following additions in the revision:
>
> **Provide a error bound between the learned K-step unrolled map vs. full LATINO iterations under mild assumptions. If relevant theorems exist, include or discuss them. Or at least analyze and discuss the finite-step error and its effect on the task.**
>
> Thank you for your comment. The convergence properties of the full LATINO algorithm can be linked to those of a discretized splitting scheme for Langevin diffusion. To make this connection precise, we have added Appendix B, which formalizes the relationship between LATINO and Langevin-based sampling. Such algorithms are typically analyzed in the asymptotic regime, where convergence to an invariant measure is established as the number of iterations approaches infinity.
>
> A key advantage of our proposed implementation of LATINO, achieved through a neural-network design via deep unfolding, is that it is explicitly fine-tuned to minimize a carefully designed training loss over only a small number of iterations. This directly addresses the practical limitation that we cannot rely on long-time averaging from an MCMC chain. Moreover, by embedding the unfolded LATINO within a reverse denoising diffusion loop, convergence can be reached with significantly fewer iterations. This is related to the fact that conditioning on x_t significantly improves the conditioning of the Langevin SDE targetted by LATINO, and by doing so, it boots its non-asymptotic convergence rate. This has been clarified in the revised manuscript.
>
> **Provide a (K,N) grid under fixed NFE showing the best allocation.**
>
> Based on the reviewer's comment, a (K, N) grid, with an independent model trained for each K, is now shown in Table 11 and Figure 10. The results are based on a new experiment described in Appendix D.1, which investigates a super-resolution inverse problem on MNIST and allows precise estimation of performance metrics through extensive repetitions and by leveraging the low-dimensional manifold structure of MNIST (MNIST digits can be reliably encoded as Gaussian parameters in a d=12 latent space, enabling a meaningful comparison between the Fréchet distances of ground-truth and reconstructed images).

---

### Decision · Action_Editor_ZDEK · 2025-11-10

**Recommendation:** Accept with minor revision

**Additional Comments:**

Currently the paper appears to be in a "diff" state with red text indicating changes. Please prepare a camera ready version of the paper.

**Audience:**

Yes

**Audience Explanation:**

All reviewers agreed that the topic and findings of the paper would be of interest to some of TMLR's audience.

**Claims And Evidence:**

Yes

**Claims Explanation:**

All reviewers agreed that the claims are supported by clear evidence.